_Article_

# The Israeli acute paralysis virus IRES captures host ribosomes by mimicking a ribosomal state with hybrid tRNAs

Francisco Acosta-Reyes[1,†], Ritam Neupane[1,2,†], Joachim Frank[1,2,*] & Israel S Fernández[1,**]

## Abstract

**Colony collapse disorder (CCD) is a multi-faceted syndrome decimating bee populations worldwide, and a group of viruses of the widely distributed Dicistroviridae family have been identified as a causing agent of CCD. This family of viruses employs non-coding RNA sequences, called internal ribosomal entry sites (IRESs), to precisely exploit the host machinery for viral protein production. Using single-particle cryo-electron microscopy (cryo-EM), we have characterized how the IRES of Israeli acute paralysis virus (IAPV) intergenic region captures and redirects translating ribosomes toward viral RNA messages. We reconstituted two _in vitro_ reactions targeting a pre-translocation and a post-translocation state of the IAPV-IRES in the ribosome, allowing us to identify six structures using image processing classification methods. From these, we reconstructed the trajectory of IAPV-IRES from the early small subunit recruitment to the final post-translocated state in the ribosome. An early commitment of IRES/ribosome complexes for global pre-translocation mimicry explains the high efficiency observed for this IRES. Efforts directed toward fighting CCD by targeting the IAPV-IRES using RNA-interference technology are underway, and the structural framework presented here may assist in further refining these approaches.**

**Keywords** internal ribosomal entry sites; Israeli acute paralysis virus; ribosome; translation

**Subject Categories** Translation & Protein Quality; Structural Biology

**The EMBO Journal (2019) 38: e102226**

## Introduction

_Apis mellifera_, the common western honey bee, is affected worldwide by an enigmatic syndrome characterized by a drastic disappearance of the workforce, causing the accelerated collapse of the hive (Ratnieks & Carreck, 2010). Given the essential role bees play

in pollination of economically important crops, the impact of this syndrome, termed colony collapse disorder (CCD), has been estimated to cost the US economy $15 billion in direct loss of crops and $75 billion in indirect losses (Chopra _et al_, 2015).

Though the exact etiology of CCD is unknown (Anderson & East, 2008), a group of viruses belonging to the _Dicistroviridae_ family were found in metagenomic studies of CCD-affected hives (Cox-Foster _et al_, 2007; Chen _et al_, 2014). Among this group of viruses, the Israeli acute paralysis virus (IAPV) showed a strong correlation with CCD, revealing a prominent role in the development of the syndrome (Hou _et al_, 2014; Doublet _et al_, 2017).

The _Dicistroviridae_ family of viruses exhibits a wide environmental distribution, targeting invertebrates, mainly insects and other arthropods (Shi _et al_, 2016). The genetic architecture of these viruses is composed of a single positive-stranded RNA molecule which contains two open reading frames (ORF1 and ORF2; Wilson _et al_, 2000b; Pisarev _et al_, 2005). ORF1 encodes non-structural proteins: an RNA helicase, a cysteine protease, and an RNA-dependent RNA polymerase (RdRP). ORF2 encodes a single poly-protein that, upon proteolytic digestion, generates the structural proteins that will eventually compose the viral capsid (Kerr & Jan, 2016; Mullapudi _et al_, 2017).

Both ORFs are preceded by non-coding RNA sequences responsible for the regulation of the expression of their downstream genes (Wilson _et al_, 2000a,b; Gross _et al_, 2017). A fine balance between the expression of ORF1 and ORF2 is required for the replication and expansion of the virus (Carrillo-Tripp _et al_, 2016; Khong _et al_, 2016). This is achieved through a precise exploitation of host resources, specially the machinery for protein synthesis (Kerr & Jan, 2016). The non-coding RNA regions preceding both ORFs harbor two different internal ribosomal entry sites (IRESs; Wilson _et al_, 2000b). IRESs are structured RNA sequences able to interfere with canonical translation, capturing host ribosomes in order to redirect them toward the production of viral proteins (Yamamoto _et al_, 2017). Eukaryotic ribosomes are operated by a complex collection of cellular factors that regulate the production of proteins in the cell (Jackson _et al_, 2010). Specially regulated in eukaryotes is the first step of translation, initiation (Aylett & Ban, 2017). During this

1 Department of Biochemistry and Molecular Biophysics, Columbia University, New York, NY, USA
2 Department of Biological Sciences, Columbia University, New York, NY, USA
*Corresponding author. Tel: +1 212 305 9512; E-mail: jf2192@cumc.columbia.edu
**Corresponding author. Tel: +1 2 2 342 2385; E-mail: isf2106@cumc.columbia.edu
[†]These authors contributed equally to this work

initiation phase, the small ribosomal subunit (40S), in partnership with many initiation factors, is able to capture an mRNA, localize its AUG initiation codon, deliver the first aminoacyl-tRNA, and finally recruit the large subunit (60S) in order to assemble an elongation competent ribosome (80S) primed with an aminoacyl-tRNA in the P site and a vacant A site (Hinnebusch & Lorsch, 2012).

The majority of IRES families leverage the complexity of initiation to hijack cellular ribosomes (Yamamoto *et al*, 2017; Jaafar & Kieft, 2019). The IAPV-IRES found in the intergenic region of the IAPV virus belongs to the well-characterized type IV family of viral IRESs. IRES sequences from this group dispense with all canonical initiation factors and are able to assemble by themselves an elongation competent ribosome, successfully redirecting the cellular machinery for viral protein production by an RNA-only mechanism (Hertz & Thompson, 2011). This is accomplished by an elaborate use of intrinsically dynamic elements of the ribosome, naturally involved in translocation (Noller *et al*, 2017a). These IRESs are able to induce an artificial state on the ribosome, mimicking a pre-translocation state with tRNAs. Elongation factors eEF2 and eEF1A can then be recruited to effectively by-pass the highly regulated initiation (Abeyrathne *et al*, 2016; Murray *et al*, 2016; Pisareva *et al*, 2018), jumpstarting directly in the elongation phase (Johnson *et al*, 2017).

The type IV IRES family exhibits a remarkable structural diversity, which remains poorly characterized (Hertz & Thompson, 2011). Two genera, based on phylogenetic analysis of ORF2 as well as the intergenic region, have been defined: Aparaviruses and Cripaviruses. The cricket paralysis virus IRES (CrPV-IRES), the prototypical Cripavirus, has been extensively studied due to its early discovery and use as model mRNA of early studies in translation (Jan *et al*, 2001; Pestova *et al*, 2004). Recently, a divergent IRES sequence of a shrimp-infecting virus, the Taura virus syndrome IRES, has been visualized by cryo-EM in complex with yeast ribosomes (Koh *et al*, 2014; Abeyrathne *et al*, 2016).

The IAPV-IRES presents the prototypical features of an Aparavirus, with an additional stem loop (SL-III) nested within the pseudoknot I (PKI) and an extended L1.1 region (Au *et al*, 2015). Importantly, this IRES can drive translation in two different ORFs, able to produce two different polypeptides from the same mRNA. A frameshift event at the first coding codon is responsible for this multi-coding capacity (Wang & Jan, 2014).

Research efforts directed toward finding the cause of CCD and developing strategies to prevent it are underway (Hunter *et al*, 2010; Chen *et al*, 2014). RNA interference has proved effective in protecting against CCD. Directing double-stranded RNAs complementary to the IRES of the intergenic region of the IAPV virus decreases the probability of hive collapse, preventing effectively the massive death of the workforce, guaranteeing the protection of the queen and thus the survival of the colony (Maori *et al*, 2009).

Using single-particle cryo-electron microscopy (cryo-EM), we have characterized how the IAPV-IRES redirects the host machinery for viral protein synthesis exploiting novel ribosomal sites. An early commitment of IRES/ribosome complexes toward global pre-translocation mimicry explains the high efficiency in ribosome hijacking observed for this IRES. These results may inspire structure-based rational designs for the fight against CCD by RNA-interference technology (Chen *et al*, 2014).

# Results

## Biochemical set-up and cryo-EM strategy

Previous biochemical and genetic studies of IAPV-IRES established the secondary structure scheme displayed in Fig 1A (Au *et al*, 2015). The prototypical architecture of the type IV IRES family consisting of three nested pseudoknots is extended by a 5′ terminal stem loop (SL-VI) proposed to play functional roles in the early positioning of the IAPV-IRES in the ribosome (Schuler *et al*, 2006; Au *et al*, 2018). Additionally, the genus Aparavirus is characterized by an extended PKI which contains an insertion of a large stem loop (SL-III, Fig 1A, bottom). In the IAPV-IRES, SL-III consists of eight Watson–Crick canonical base pairs and a terminal loop of six nucleotides. Notably, two unpaired adenine residues are placed in a strategic position at the core of the three-way helical junction connecting SL-III, the anti-codon stem loop (ASL)-like element of the PKI (residues 6,546–6,574), and the double-helical segment connecting PKI and PKIII. A variable loop region (VLR) bridges the mRNA-like element of PKI (residues 6,613–6,617) with the helical region connecting PKI and PKIII. This single-stranded RNA loop is poorly conserved in sequence; however, even though its role in IRES functioning remains enigmatic, biochemical experiments have proved its integrity is mandatory for productive IRES-driven translation (Ruehle *et al*, 2015).

In order to understand in structural terms how these constituent units of the IAPV-IRES are involved in ribosome hijacking, we produced a full, wild-type IAPV-IRES, including SL-VI and the first two coding codons. Binary complexes with mammalian ribosomes and IAPV-IRES were generated by incubating IRES with ribosomal subunits. We designed a reaction featuring an excess of 40S over 60S in an overall background of IRES excess, to test the ability of the IAPV-IRES to engage both 40S and full 80S ribosomes in a productive and stable binary interaction. The stability of these interactions was tested through a sucrose gradient run overnight (Fig 1B) where the different complexes could be resolved according to their size differences. Each peak was subjected to RNA extraction and UREA-PAGE analysis where the presence of IAPV-IRES bound in the 80S peak as well as in the 40S peak could be confirmed (Fig 1B).

Leveraging latest cryo-EM maximum-likelihood classification methods implemented in RELION 3.0 (Scheres, 2012; von Loeffelholz *et al*, 2017; Zivanov *et al*, 2018), we decided to directly image the above reaction without the sucrose gradient step. A large dataset ensuing from this experiment was subjected to an optimized classification scheme combining different *in silico* classification approaches (Appendix Fig S1). This allowed us to identify and refine to high-resolution five distinctive classes from a single dataset (Fig 1C–E). The nominal resolution of the maps was calculated to be around 3 Å (Appendix Figs S2, S3, and S6). In the best areas, such as the 60S subunit and the body of the 40S subunit, the maps exhibit characteristics in accordance with this resolution, with very well-resolved side chains in proteins and clear base separation in the ribosomal RNA components (Appendix Fig S4A and B). However, the resolution of the IRES density, due to the intrinsic flexibility of this component, deviates from the nominal resolution. Areas of the IRES stabilized by ribosomal components are well-resolved, with local resolution better than 4 Å (Fig 3D and E), while

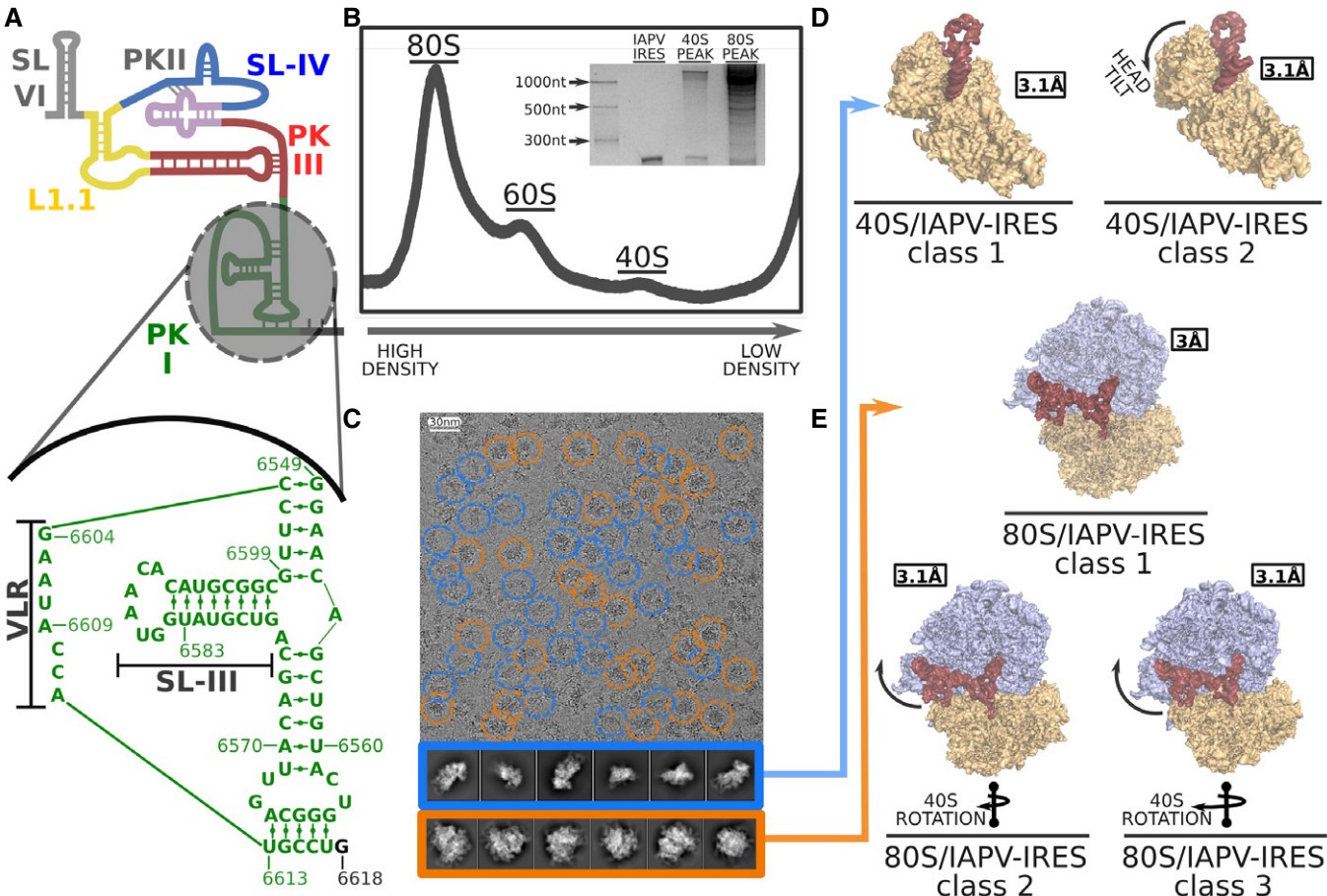

**Figure 1. IAPV-IRES secondary structure, experimental set-up, and cryo-EM image processing workflow.**

A   IAPV-IRES diagram colored according to secondary structure motifs. Bottom, a closer view of the IAPV-IRES PKI highlighting its sequence, with base pairs indicated as well as the variable loop region (VLR) and stem loop III (SL-III).

B   Sucrose gradient UV profile of 60S/40S/IAPV-IRES reaction mixture after an overnight run. The peaks corresponding to 80S and 40S were used for RNA extraction and UREA-PAGE shown in the inset.

C   Representative cryo-EM image where roughly half of the particles correspond to 40S (blue) and the other half to 80S (orange).

D   Two classes with robust density for the IAPV-IRES were found in the 40S group.

E   After classification, three classes with clear IAPV-IRES density and small differences in the conformation of the 40S were found in the 80S group.

areas not stabilized by the ribosome or in contact with intrinsically dynamic elements of the ribosome like the L1 stalk exhibit lower local resolution (Appendix Figs S2, S3, and S6). In order to properly visualize the continuity of the maps for the full IRES, we show the unsharpened maps in the figures, especially where large areas of the maps are depicted. In those regions of the maps exhibiting resolution better than 4 Å for the IRES, maps sharpened with B factors reported in Appendix Table S1are shown.

**The IAPV-IRES restricts the conformational freedom of the 40S blocking functional sites**

The 40S subunit can be roughly divided into two parts: the body, which forms the bulk of the subunit accounting for two-thirds of its mass, and a more mobile part roughly comprising the remaining third, designated as the head (Fig 2A). The interface between these two components forms the tRNA binding sites of the small subunit.

The head of the 40S subunit is a dynamic component, modifying its relative orientation with respect to the body. This dynamics is of critical importance in two aspects of translation: the positioning of the initiator aminoacyl-tRNA and in the concerted movement of mRNA and tRNAs during elongation (Ramrath *et al*, 2013; Llacer *et al*, 2015). We identified two classes of particles showing robust density for IAPV-IRES in the context of a binary interaction with the 40S (Fig 2B and C). All elements of the IAPV-IRES included in the produced construct were identified in the maps except SL-VI, which proved disordered—no density could be assigned to it even in low-pass filtered maps. The L1.1 region in the context of a binary interaction with the 40S shows a high degree of mobility and can only be modeled in maps filtered to 4 Å.

The IAPV-IRES inserts two elements of its structure between the head and the body of the 40S subunit, effectively restricting the dynamics of the 40S head to specific ranges of conformations. The ASL/mRNA mimicking part of the PKI is inserted in the decoding

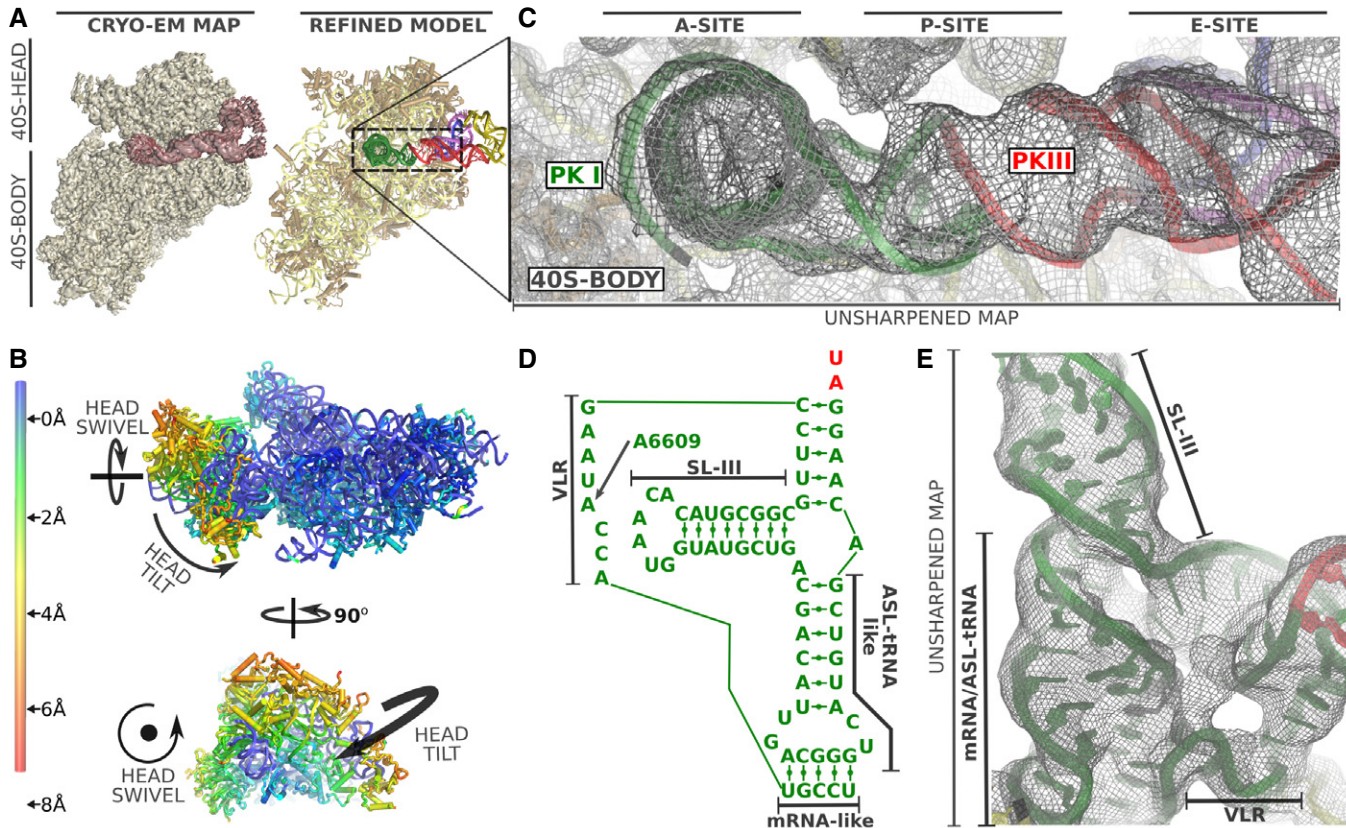

**Figure 2. Structure of the IAPV-IRES in complex with the 40S ribosomal subunit.**

A  Overview of the mammalian 40S in complex with IAPV-IRES. Left, cryo-EM final post-processed map of class 1 with 40S colored yellow and IAPV-IRES maroon. Right, corresponding final refined model with IAPV-IRES colored according to Fig 1A.

B  Ribbon diagram of the 40S colored by pairwise root-mean-square deviation displacements observed between the two IAPV-IRES/40S classes. The different position of the 40S head between both classes is a composition of swiveling and tilt movements (indicated by arrows in orthogonal views).

C  Close-up view of the ribosomal sites of the 40S for IAPV-IRES/40S class 1 showing cryo-EM unsharpened cryo-EM density.

D  Sequence of the PKI three-way helical junction.

E  Unsharpened cryo-EM density for the PKI region of the IAPV-IRES in class 1 with the SL-III and the tRNA/mRNA mimicking domain indicated.

site (A site) of the small subunit stabilized by the decoding bases of the 18S rRNA A1824-A1825 and G626 [A1492, A1494, and G530 in *Escherichia coli* (Ogle & Ramakrishnan, 2005)], inducing a decoding event. SL-IV is deeply inserted in the interface of head and body, in the surroundings of the E site, clamped by stacking interactions established with residue A6498 of the IRES and tyrosine 72 from uS7 and arginine 135 from uS11 (Fig 3E). In this conformation, the IAPV-IRES fully blocks all three tRNA binding sites of the 40S subunit, interfering with early steps of canonical initiation (Fig 2C; Aylett & Ban, 2017).

Density for the full PKI, including SL-III, was clearly visible in the maps (Fig 2D and E, and Appendix S3) which allowed accurate modeling of the three-way helical junction characteristic of Aparavirus IRESs. The VLR was also visible in the maps, partially occupying the P site, stabilized by a stacking interaction between A6609 of the IRES and A1085 of the 18S rRNA (Fig 2E).

The two classes present a conformation of the IAPV-IRES nearly identical (r.m.s.d. = 1.12 Å between the two IRES conformations) but in displaced position with respect to the 40S body (Fig 3A and

B). The IAPV-IRES seems to follow the movement of the 40S head, pivoting around the anchored PKI and SL-IV which are exceptionally stabilized by ribosomal elements from both head and body, effectively "clamping" the IRES to the 40S subunit (Fig 3D and E).

The three-way helical junction modeled in the PKI of the IAPV-IRES resembles a "hammer" shape, with SL-III coaxially stacked on top of the ASL-like stem (Fig 2D and E). Perpendicular to both and situated in between them, a helical segment connects PKI and PKIII. The coaxially aligned SL-III and ASL-like domain forms a straight unit of shape and dimensions similar to a tRNA, excluding the acceptor stem (Fig 3C). Alignments of structures containing tRNAs in several configurations [canonical tRNA PDBID:4V5D (Voorhees *et al*, 2009), with A/T-tRNA PDBID:5LZS (Shao *et al*, 2016) and hybrid tRNA PDBID:3J7R (Voorhees *et al*, 2014)] with the structure of the IAPV-IRES in complex with the 40S, reveal an interesting positioning of the coaxial unit formed by SL-III and the ASL-like part of PKI (Fig 3C). Notably, the SL-III/ASL-like unit of IAPV-IRES populates a space more similar to a hybrid A/P-tRNA than a canonical, A/A-tRNA or A/T-tRNA [following nomenclature of hybrid

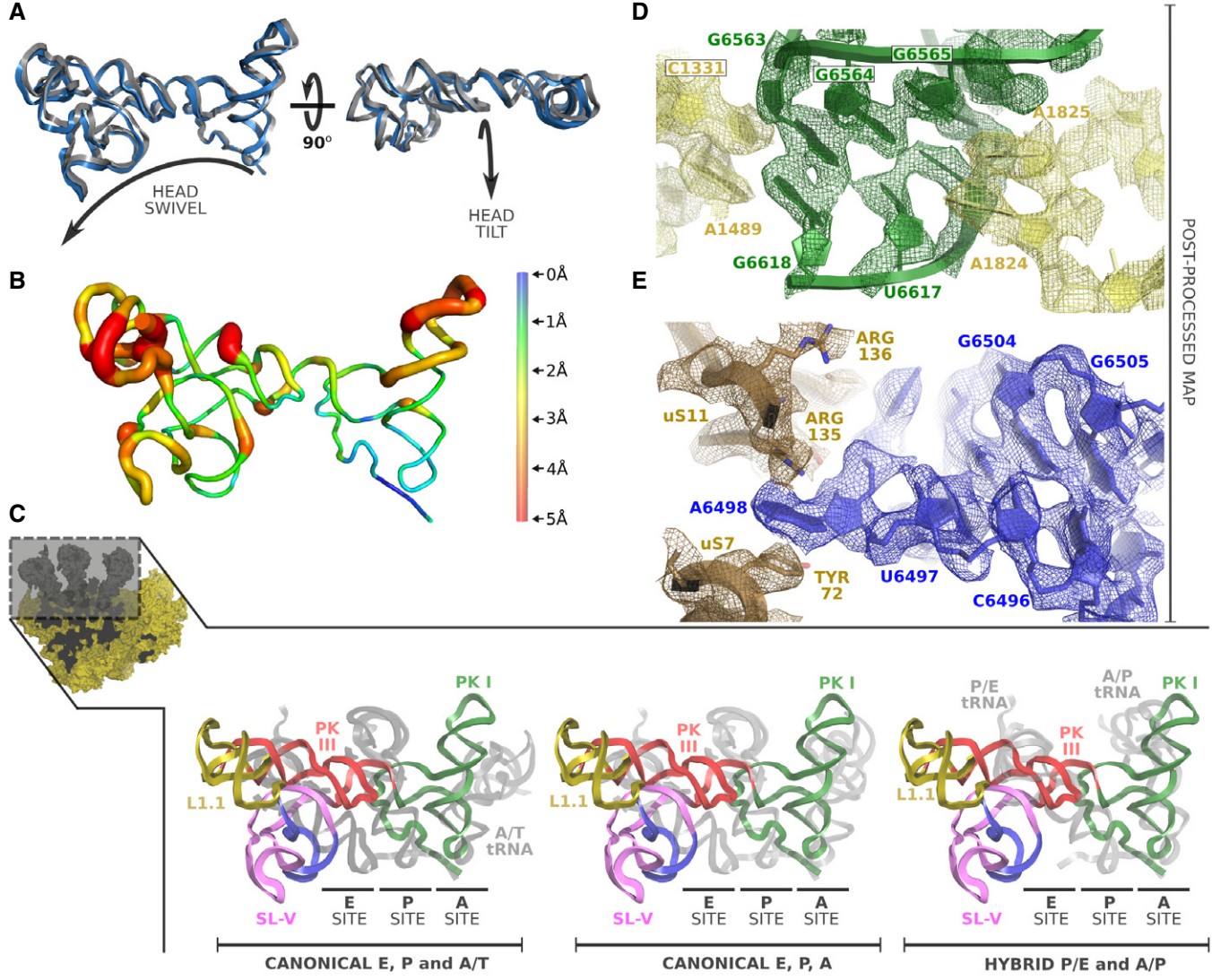

**Figure 3. IAPV-IRES conformation in the context of a 40S interaction.**

A Superposition of IAPV-IRES models corresponding to IAPV-IRES/40S class 1 and class 2 after alignments excluding the IRES and the 40S head. A similar conformation can be observed with distinctive relative orientation with respect to the 40S body. The movement characteristic of the 40S head is indicated by arrows in orthogonal views.

B Ribbon diagram of the IAPV-IRES colored by pairwise root-mean-square deviation displacements observed between the two IAPV-IRES/40S classes. The ASL/mRNA-like regions of the PKI and well as the SL-IV show the lowest degree of displacement (blue), whereas the apical part of SL-III and the L1.1 the highest (red).

C Superposition of the IAPV-IRES with tRNAs in different configurations indicated at the bottom. IAPV-IRES is depicted as ribbons colored according to the secondary structure elements, and tRNAs are represented as gray ribbons. Alignments of the models were computed with the 40S body, excluding from the computation the ligands (IRES/tRNAs) and the 40S head.

D Detailed view of the refined model for IAPV-IRES/40S class 1 inserted in the post-processed cryo-EM density focused on the decoding center of the 40S. PKI of IAPV-IRES is depicted green and 18S rRNA yellow.

E Close-up view of the refined model for IAPV-IRES/40S class 1 inserted in the post-processed cryo-EM density focused on the SL-IV of the IAPV-IRES (depicted blue).

states previously proposed (Ratje *et al*, 2010)]. Similarly, PKIII overlaps with the position occupied by a hybrid P/E-tRNA, mimicking its helical components of the elbow region of a tRNA in this intermediate configuration. Overall, the IAPV-IRES is able to manipulate the 40S subunit in isolation, blocking the functional sites where canonical initiation factors eIF1, eIF1A, and eIF5B bind and, at the same time, steering the intrinsic dynamics of the 40S head toward a configuration reminiscent of an early elongation, pre-translocated state.

**SL-III interacts with ribosomal protein uL16 stabilizing the 80S in a pre-translocation configuration mimicking hybrid tRNAs**

The binary IAPV-IRES/80S complex populates three major conformations, with limited differences between them (Fig 1E). The majority of particles populated a class where the 40S subunit exhibits a small degree of intersubunit rotation (approx. 1°) compared with the unrotated, canonical configuration (Fig 5A). No major

swiveling or tilt of the 40S head is visible in this conformation. The IAPV-IRES maintains a similar global conformation as in the binary complex with 40S, but in the 80S map, both the L1.1 region and the tip of the SL-III show good density as their dynamics are restricted by specific contacts with elements of the 60S: The L1 stalk stabilizes the L1.1 region and the A site finger and the ribosomal protein uL16 the SL-III. The A site finger (28S rRNA helix 38) is a flexible component of the 28S rRNA which plays an important role in translocation of tRNAs from the A to the P site (Nguyen *et al*, 2017). In many structures, it is not visible due to its intrinsic flexibility, required to perform its role escorting in-transit tRNAs (Brown *et al*, 2016; Nguyen *et al*, 2017). The SL-III of IAPV-IRES contacts the A site finger, stabilizing it in a fixed conformation, which allows the apical loop of SL-III to reach deep into the 60S, establishing a novel interaction with the ribosomal protein uL16 (Fig 4A and B). The IAPV-IRES positions the apical loop of SL-III (nucleotides 6,585–6,590) in direct contact with basic residues of uL16, which are in electrostatic interacting distance with negatively charged phosphates of the RNA backbone of the IRES (Fig 4C). The additional anchoring points to the ribosome contributed by SL-III, allows the IAPV-IRES, in the context of an 80S interaction, to be stabilized in a conformation that overlaps with the space occupied by a hybrid A/P-tRNA. The coaxial unit SL-III/ASL-like domain of the IAPV-IRES functionally mimics an A/P-tRNA priming the 80S for eEF2 recruitment, effectively bypassing the initiation stage (Fig 4D). Additionally, the anchoring points provided by the IAPV-IRES along the intersubunit space

probably contribute to an effective recruitment of the 60S in the absence of the dedicated factor responsible for such function in canonical translation, eIF5B (Pestova *et al*, 2000).

## Aparavirus IRESs restrict the small subunit rotation dynamics in the pre-translocation state

No populations with wide rotations of the small subunit were identified in our large 80S/IAPV-IRES dataset. This suggests that, in contrast with Cripavirus IRESs (Fernandez *et al*, 2014; Koh *et al*, 2014), the IAPV-IRES is able to restrict the dynamics of the small subunit, channeling it toward a canonical, non-rotated configuration (Fig 5A). This is accomplished by a solid anchoring of the PKI in the A site, which not only mimics the ASL of a tRNA interacting with a cognate codon in the A site, but also, by placing the SL-III in a similar position as a hybrid A/P-tRNA (Moazed & Noller, 1989; Frank, 2012), mimics the T and D arms of a tRNA (Figs 4D and 5B, and Appendix Fig S7). In such position, the PKI of the IAPV-IRES establishes a network of interactions with both the large and the small subunits, effectively restricting the rotation of the 40S. Apart from the interactions established by the apical loop of the SL-III with ribosomal protein uL16 (Fig 4C), the decoding event elicited by the placement of the PKI in the decoding center allows the establishment of an interaction with the 28S rRNA base A3760 (A1913 in *E. coli*), normally involved in decoding (Fig 5C; Demeshkina *et al*, 2010). This interaction is maintained along the

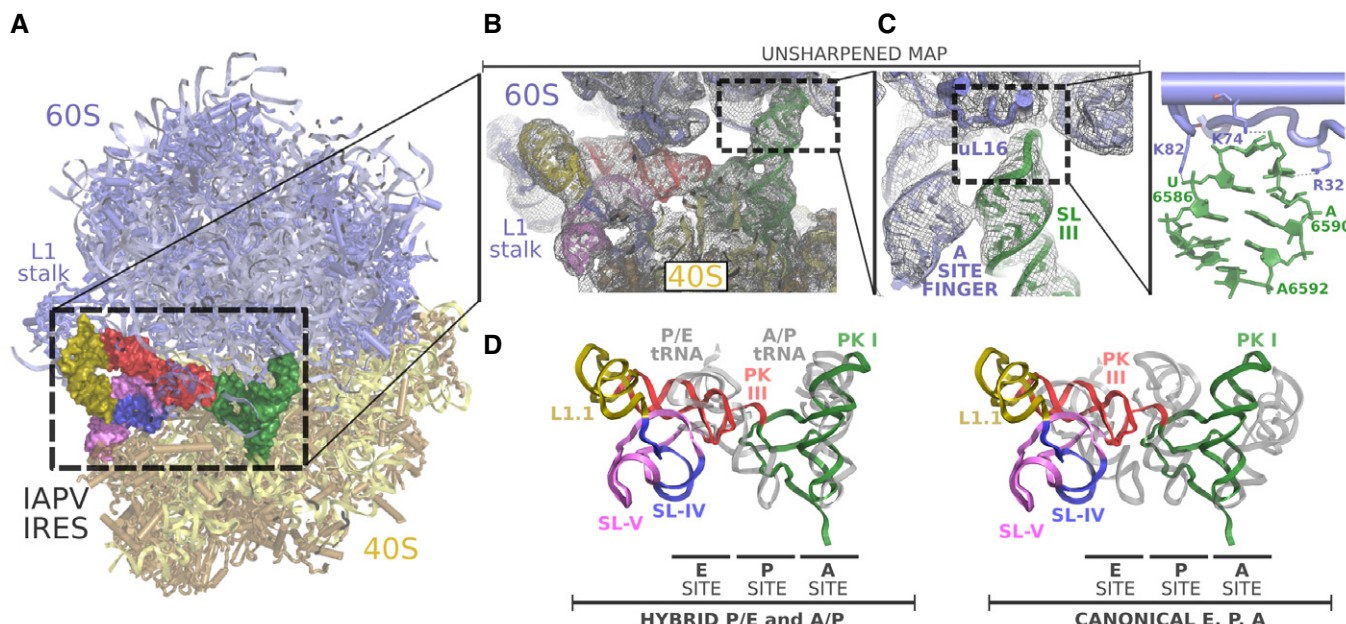

**Figure 4.  SL-III of IAPV-IRES engages novel sites of the 60S ribosomal subunit.**

A   Overall view of the IAPV-IRES/80S complex class 1 with 60S represented as cyan ribbons, the 40S as yellow ribbons, and the IAPV-IRES represented as solid Van der Waals surface colored by secondary structure motifs.

B   Close-up view of the intersubunit space with the IAPV-IRES depicted as cartoons colored as in (A) inserted in the unsharpened cryo-EM density.

C   Zoomed view of the A site finger in interacting distance with the SL-III (green). The apical loop of SL-III reaches deep into the 60S contacting the ribosomal protein uL16.

D   Superposition of the IAPV-IRES in complex with 80S (class 1) with tRNAs in different configurations indicated at the bottom. Alignments of the models were computed with the 40S body, excluding from the computation the ligands (IRES/tRNAs) and the 40S head. IAPV-IRES PKI component SL-III/ASL-like domain populates a space of the intersubunit space similar to a A/P-tRNA. IAPV-IRES PKIII (red) mimics the elbow region of a hybrid P/E-tRNA.

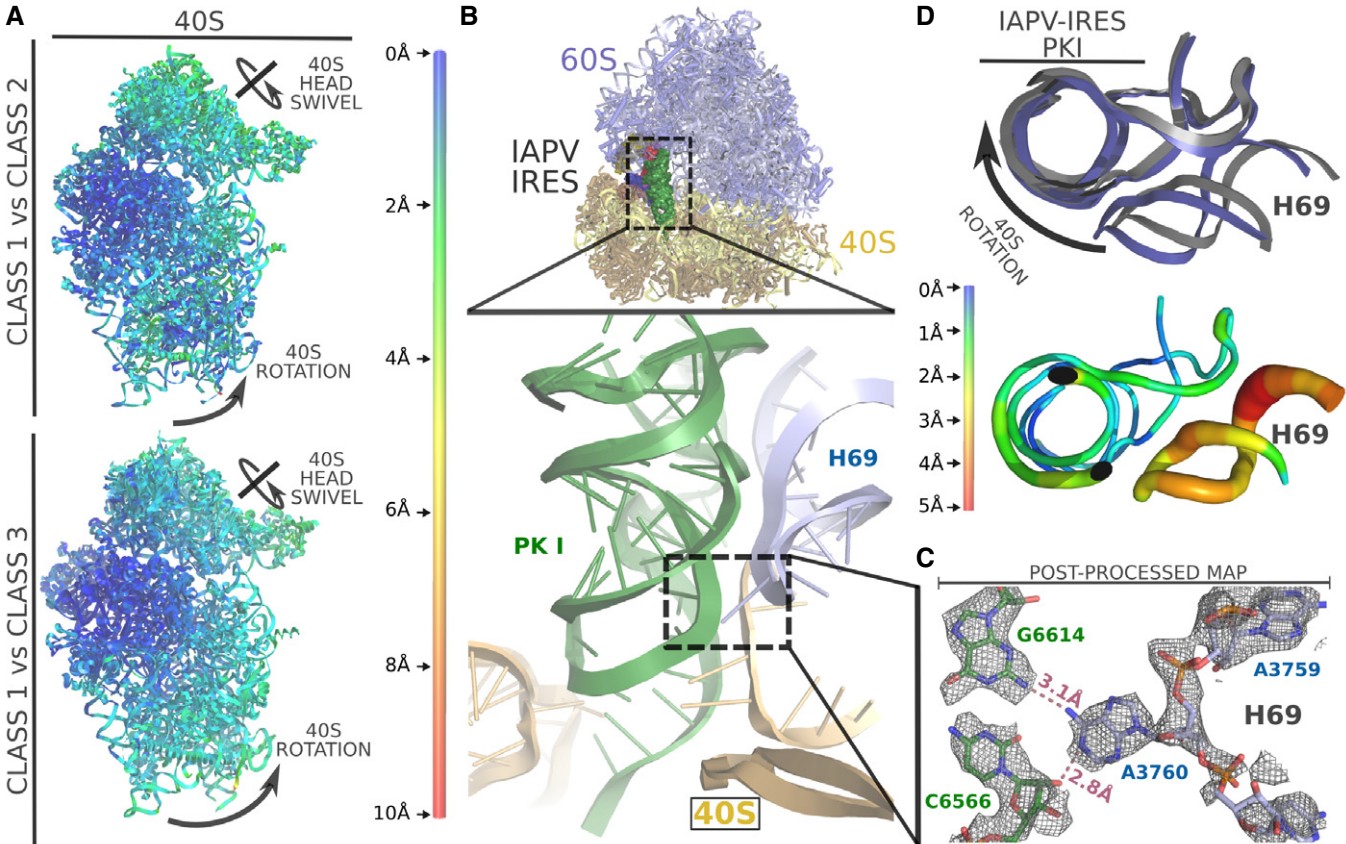

**Figure 5.  The IAPV-IRES restricts the small subunit rotational dynamics in a pre-translocation complex with 80S.**

A   Ribbon diagram of IAPV-IRES/80S complex viewed from the 40S colored by pairwise root-mean-square deviation displacements observed between the classes indicated on the left. Class 1 is unrotated, while classes 2 and 3 exhibit a small rotational movement of the 40S.

B   Top, general overview of the non-rotated IAPV-IRES/80S class 1 structure. IAPV-IRES is depicted as solid Van Der Waals surface colored according to the secondary structure motifs. The PKI (green) is solidly anchored to the A site. Bottom, close-up view of the IAPV-IRES PKI inserted in unsharpened map.

C   Zoomed view of A3760, a nucleotide belonging to the helix 69 (H69) of the 28S rRNA interacting with PKI. Final refined model inserted in the post-processed map is shown.

D   This interaction is not disrupted along the small fluctuations of the 40S. An apical view along the axis of the PKI of a superposition of class 1 versus class 2 shows the IRES displacements are minimal and are followed by the H69 which constantly interacts with the IRES.

small fluctuations of the 40S, which, locally, are restricted to displacements of a few Angströms (Fig 5D). The IAPV-IRES seems to bind very tightly in a binary, pre-translocation complex with the 80S. The recruitment of elongation factors to commit the ribosome to the production of viral proteins seems to be achieved not through a dynamic manipulation of the 40S rotation, but by directly adopting a configuration reminiscent of a ribosome with tRNAs in hybrid configurations.

**Remodeling of specific components of the IAPV-IRES allows its translocation through the ribosome**

Due to their intrinsic flexibility, IRESs are able to populate multiple conformational states, while maintaining a basic structural framework dictated by their base-pairing scheme. A combination of rigid elements connected via flexible linkers allows these RNAs to tune their interactions with different ribosomal sites as they transit from an early pre-translocated state to a post-translocated one. Along this

vectorial movement, these IRESs take advantage of intrinsic dynamic elements of the ribosome, normally involved in translocation of tRNAs and mRNAs (Voorhees & Ramakrishnan, 2013; Noller et al, 2017a).

In order to visualize the IAPV-IRES in a post-translocated state, we engineered a stop codon in the first coding codon following the IRES sequence (Muhs et al, 2015; Pisareva et al, 2018). By simultaneously incubating a pre-translocated 80S/IAPV-IRES complex with eEF2 and a catalytically inactive version of the eukaryotic release factor 1 (eRF1*) (Alkalaeva et al, 2006) in the presence of GTP, we were able to stabilize the IRES after a single translocation on the ribosome, allowing the visualization of the overall conformation of the IRES in a post-translocated state as well as the specific determinants of the IRES in binding the ribosomal P site (Fig 6A).

Globally, the post-translocated IAPV-IRES exhibits an extended conformation with the ASL/SL-III unit deeply inserted in the P site and the L1.1 region maintaining its original connection with the L1 stalk. The SL-IV and SL-V of the IRES are no longer in contact with

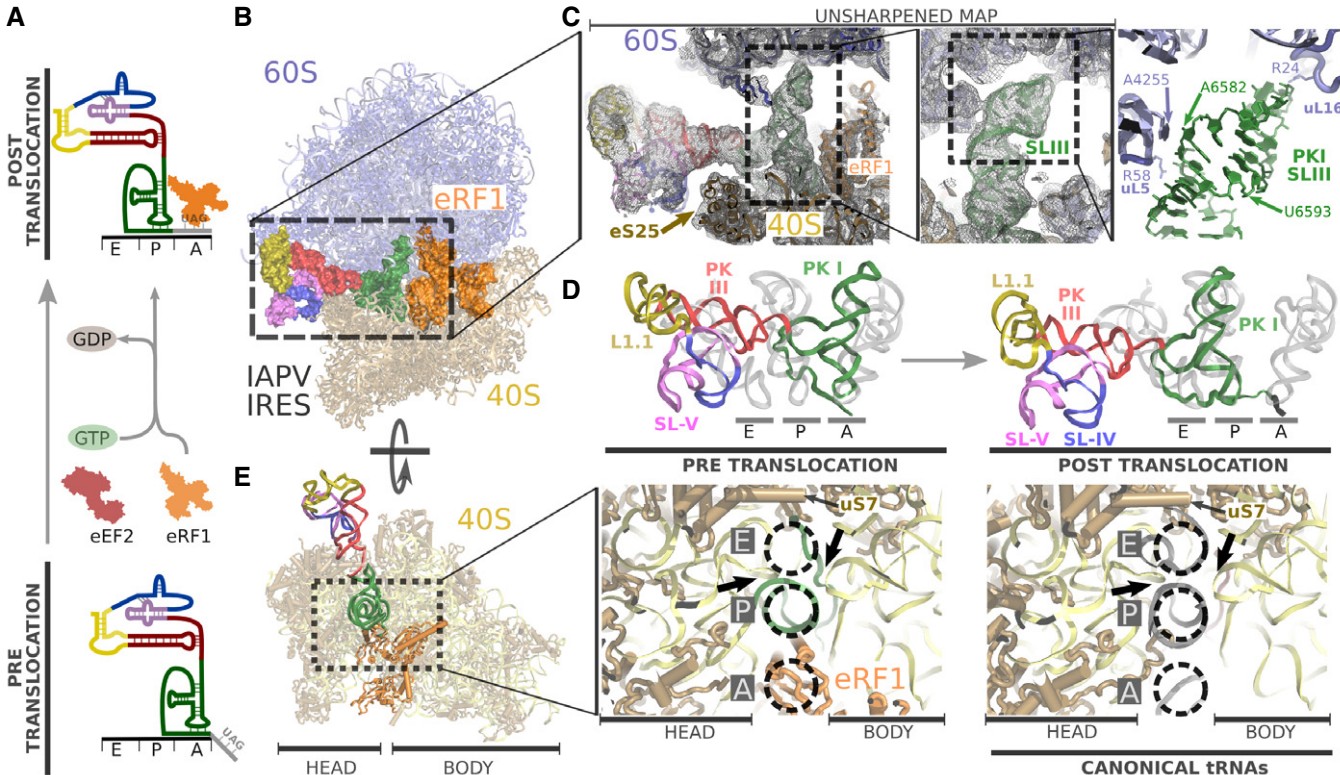

**Figure 6. Visualization of IAPV-IRES in a post-translocated state in the ribosome.**

A Biochemical strategy employed to trap a post-translocated state of IAPV-IRES in the ribosome.

B General view of the IAPV-IRES in a post-translocated state on the ribosome: 60S depicted as blue ribbons, 40S as yellow ribbons, IAPV-IRES represented as solid Van Der Waals surface colored according to secondary structure elements described in Fig 1A, and eRF1* depicted orange.

C Close-up view of the SL-III inserted in the experimental unsharpened cryo-EM density. On the right, refined model with residues from the 60S (blue) in interacting distance with the SL-III (green) indicated.

D Comparison of the final refined model for IAPV-IRES colored according to the secondary structure described in Fig 1A with canonical tRNAs (PDBID: 4V5D) in the pre-translocated state (left) and after translocation (right).

E Left, overall top view of the intersubunit space of the 40S for the post-translocated state. Inset, close-up view of the 40S tRNA binding sites where it can be appreciated the insertion of PKI of the IAPV-IRES in the P site, projecting the VLR toward the E site. The elements of the 18S rRNA forming the "P site gate" are indicated by solid arrows. On the right, equivalent view for canonical tRNAs.

the 40S (Fig 6B and C left). The SL-III seems to play an important role in orienting the IRES unit formed by the ASL/SL-III to a position that perfectly matches that of a canonical P site tRNA (Fig 6D). Nucleotides belonging to the SL-III establish contacts with several residues of ribosomal proteins uL5 and uL16 as well as with the 28S rRNA nucleotide A4255, all components of the large ribosomal subunit (60S) (Fig 6C, right).

In canonical translation, the movement of tRNAs and mRNA has to be coordinated in order to vacate the ribosomal A site for the next incoming aminoacyl-tRNA. After peptidyl transfer, the ribosome adopts a rotated configuration of the small ribosomal subunit with tRNAs in hybrid configurations. The movement of the peptidyl-tRNA in the A site to the P site has to be coordinated with the movement of the P site tRNA to the E site, and the tRNA occupying the E site has to be ejected from the ribosome with the assistance of the L1 stalk. During this process, it is of capital importance that the correct reading frame on the mRNA be maintained (Voorhees & Ramakrishnan, 2013; Noller *et al*, 2017a,b). This is accomplished by the participation of specific components of

the ribosome (mainly RNA bases) which interact with tRNAs and mRNA defining specific checkpoints so as to prevent in-transit tRNAs from slipping or loosing contact with the mRNA on the correct frame (Zhou *et al*, 2014). One of the most important checkpoints is defined by the so-called "P site gate" (Zhou *et al*, 2014): a constriction formed by elements of the 18S rRNA (bases 1,054–1,064 and 1,638–1,645, 1,335–1,344, and 785–795 in *E. coli*) that physically block the progression of a translocated peptidyl-tRNA in the P site from slipping into the E site. The "closing" of the P site gate marks the end of a correct translocation cycle, allowing the ribosome to reset to a canonical, non-rotated configuration (Noller *et al*, 2017b).

By perfectly mimicking a canonical P site tRNA, the IAPV-IRES in a post-translocated state is able to position the VLR, a flexible element of its structure, in contacting distance with the E site of the 40S subunit. This is accomplished even with a fully closed P site gate (Fig 6E): Sliding through the P site gate, the single-stranded VLR can contact the ribosomal protein uS7, normally involved in stabilizing E site tRNAs (Fig 6E; Brown *et al*, 2018). The VLR thus

## Cripaviruses

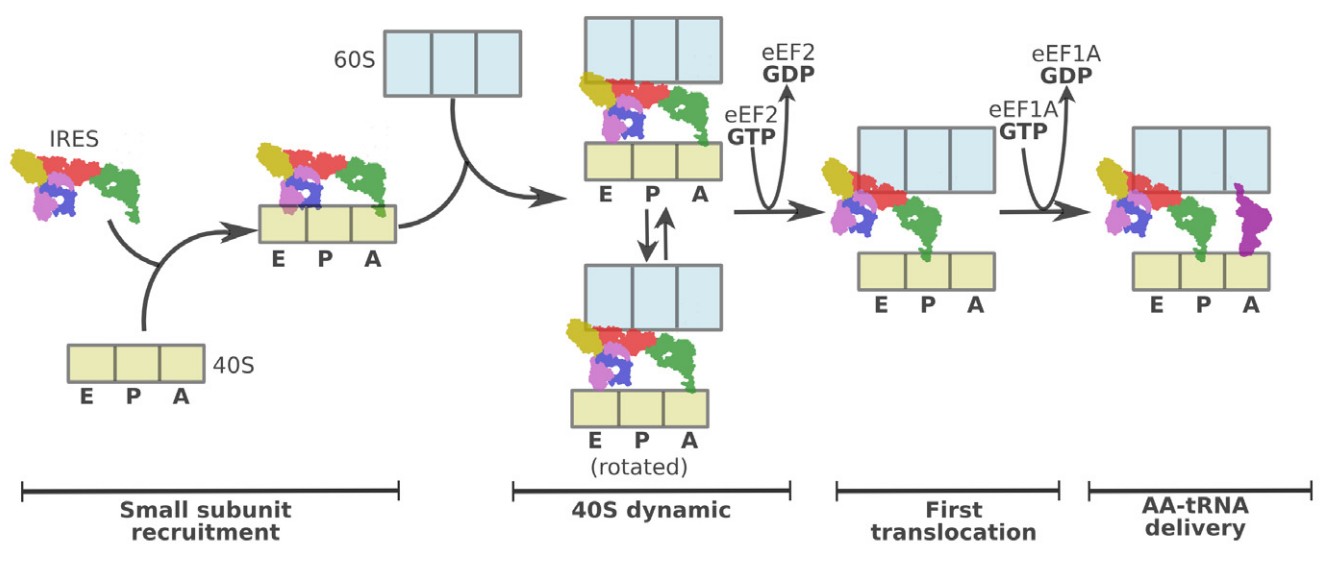

## Aparaviruses

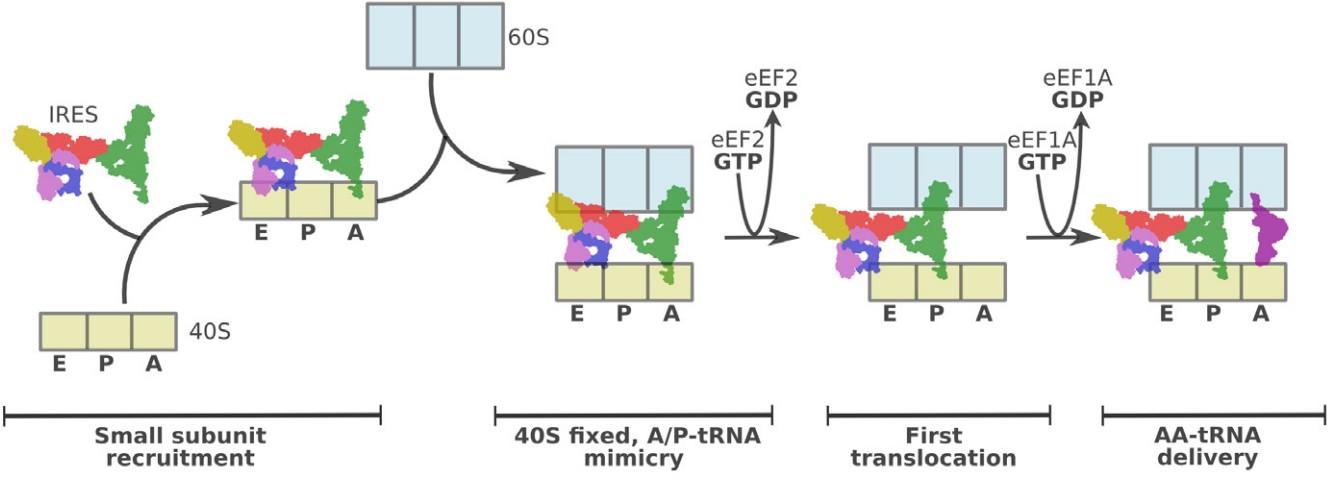

**Figure 7.    Type IV IRES families exploit different pre-translocation features of canonical translation for ribosome hijacking.**

(Top), The Cripavirus family of type IV IRESs is able to capture free 40S subunits and engage them on a pre-translocation complex by recruiting 60S. This complex is extremely dynamic, with the 40S alternating between non-rotated and rotated configurations with respect to the 60S subunit. IRESs belonging to this family, exemplified by the CrPV-IRES, recruit elongation factors by mimicking a rotated stated of the ribosome with tRNAs. (Bottom), The Aparavirus family of IRESs follows a similar pathway in order to assemble a pre-translocation complex; however, specific structural components of this family allow for additional contacts with the 60S, limiting the rotational freedom of the 40S. Elongation factor engagement and thus effective ribosome hijacking are accomplished by mimicking a ribosome state with tRNAs in hybrid configurations.

undergoes a marked remodeling as the IAPV-IRES transitions from the pre-translocation to the post-translocation state. While in the pre-translocation conformation, the VLR establishes interactions with ribosomal components of the P site, after translocation, it populates a more extended conformation that is able to reach the E site even with the P site gate locked. The remodeling of this IRES component seems to be crucial for proper translocation of the IRES, as mutations that alter both its length and its base composition impact negatively on the ability of the IRES to initiate translation (Ruehle *et al*, 2015).

## Discussion

Metagenomic studies of environmental samples have recently underscored the pervasive role RNA viruses exert on the biosphere (Shi *et al*, 2016). With estimates of dozens of RNA viruses infecting a single species, the diversity and impact these molecular entities have in biology and evolution is highly under-appreciated (Koonin & Dolja, 2013). It has been discovered that insects and arthropods host the highest diversity of RNA viruses (Shi *et al*, 2016). Within that realm, viruses from the *Dicistrovirideae* family have generated

special interest given their wide range and distribution of hosts (Cox-Foster *et al*, 2007; Shi *et al*, 2016).

Using cryo-EM, we comprehensively characterized how the intergenic IRES of IAPV, a virus of the *Dicistrivirideae* family and a causing agent of the colony collapse disorder (CCD), binds and manipulates ribosomes to redirect them toward the production of viral proteins. The IAPV-IRES is able to establish a stable binary interaction with the small ribosomal subunit. An early capturing of free 40S subunits, committing them toward viral protein production, may represent a limiting step in a cellular environment where cellular and viral messages contend for ribosomal access. The IAPV-IRES is able to insert its PKI domain in the A site (decoding site) of the 40S effectively blocking the binding of eIF1/eIF1A, two initiation factors required for canonical initiation (Fig 2C). While blocking the 40S functional sites, the IAPV-IRES is able to steer the intrinsic dynamics of the 40S subunit toward a specific configuration, facilitating the recruitment of the large ribosomal subunit (60S) in the absence of eIF5B, the cellular factor catalyzing this event.

Bypassing the highly regulated initiation stage of translation is a customary requirement for the type IV family of IRES (Jackson *et al*, 2010). This is accomplished by inducing an altered ribosome state which is able to recruit elongation factors (eEF2 and eEF1A) directly and without tRNAs. Cripavirus IRESs such as the CrPV-IRES accomplish this by inducing a wide rotation on the 40S, mimicking a pre-translocation state of the ribosome with tRNAs (Fig 7, top). In marked contrast, Aparavirus IRESs such as the IAPV-IRES capture elongation factors not by inducing a rotated state of the 40S, but by directly mimicking a ribosomal state with hybrid tRNAs (Fig 7, bottom). The limited dynamics of 40S subunit rotation observed in our large cryo-EM dataset reflects a solid anchoring of both ribosomal subunits, mediated mainly by the additional contacts contributed by the SL-III characteristic of Aparavirus IRESs. However, a compromise between rigidity and flexibility is required for these IRESs to operate: In order to place the first coding codon in the decoding center of the 40S, the IAPV-IRES has to move (translocate) from the A site to the P site. We were able to gain structural information of this transition visualizing the post-translocated state of the IAPV-IRES. In the post-translocated state, the PKI mimics a canonical P site tRNA and the SL-III establishes new contacts with proteins and rRNA components of the P site of the 60S (Fig 6C). Additionally, the extremely dynamics of a single-stranded region of the IAPV-IRES termed the VLR (Fig 2D) is able to modify its configuration in a context-specific manner. In the context of a binary interaction with the 40S subunit and in the pre-translocation state with the 80S ribosome, the VLR exhibits a compact configuration, contacting bases of the 40S subunit's P site. After translocation, and once PKI is displaced to the P site, the VLR is extended, contacting ribosomal protein uS7, a component of the 40S subunit's E site (Fig 6E). Biochemical evidence supports a key role of the VLR in IRES function, as mutations and/or shortening impacts the ability of the IRES to efficiently translocate (Ruehle *et al*, 2015).

Using high-resolution single-particle cryo-EM analysis, we show how the prototypical Aparavirus IRES of the intergenic region of the IAPV manipulates the eukaryotic ribosome to position a viral non-AUG codon in the ribosomal A site to effectively hijack the host machinery for protein production (Movie EV1). We have

uncovered a new strategy of early pre-translocation mimicry used by this IRES sub-family as well as visualize the conformational dynamics of a strategic single-stranded segment of the IRES, the VLR. The structures presented here will allow structure-based design of new and better RNA-interfering strategies directed toward the vital intergenic IRES of the IAPV. Ongoing efforts in that direction have already proved successful in fighting CCD, protecting bee hives from collapse (Maori *et al*, 2009; Piot *et al*, 2015).

# Materials and Methods

### Plasmids

Expression vector for His-tagged eRF1*(AGQ mutant) has been previously described (Ratnieks & Carreck, 2010). A pUC19-based transcription vector for IAPV-IRES-WT was constructed by inserting a T7 promoter sequence upstream of IAPV IGR IRES sequence followed by the first two coding triplets (nucleotides 6,372–6,623 from NC009025). An EcoRI site was included after the second codon. Site-directed mutagenesis was employed to change the first coding codon (GGC) to a stop codon (TAG) to create the IAPV-IRES-STOP construct. Both IRES constructs were transcribed using T7 RNA polymerase. Briefly, 0.1 mg/ml of EcoRI-linearized vector was transcribed using 0.1 mg/ml of homemade T7 RNA polymerase in 10 ml transcription buffer (100 mM HEPES-KOH pH 7.4, 10 mM of each NTP, 22 mM $MgCl_2$, 50 mM DTT, 2 mM Spermidine, 1 μl/ml IPP) for 4 h at 37°C. The RNA was then washed, concentrated, and separated on a 6% UREA-PAGE gel. The IAPV-IRES band was cut from the gel, electro-eluted, buffer exchanged into Buffer A (20 mM Tris–HCl, pH 7.5, 100 mM KCl, 8 mM $MgCl_2$, 2 mM DTT), and snap-frozen in liquid nitrogen.

### Purification of translation components

Native 40S and 60S ribosomal subunits and eukaryotic elongation factor 2 (eEF2) were prepared from rabbit reticulocytes as previously described (Pisarev *et al*, 2010). Recombinant eRF1* was purified according to a previously described protocol (Alkalaeva *et al*, 2006).

### Assembly of ribosomal complexes

To reconstitute ribosomal complexes in the pre-translocation state with IAPV-IRES, we incubated 6 pmol of 40S ribosomal subunits with 60 pmol of IAPV-IRES-WT RNA in a 15 μl reaction mixture in Buffer A for 5 min at 37°C. Then, the reaction mixture was supplemented with 4.5 pmol of 60S ribosomal subunits and incubated for 5 min at 37°C. We maintained an excess of 40S ribosomal subunits to trap both the 40S/IAPV-IRES and the 80S/IAPV-IRES complexes in the same reaction. Assembly of the complexes was verified by running the reaction through an overnight sucrose gradient, phenol extracting the RNA, and running a UREA-PAGE gel using standard methods. The assembly and integrity of the complex was also verified on a F20 screening microscope via negative stain EM and cryo-EM.

To reconstitute ribosomal complexes in a post-translocated state with IAPV-IRES-STOP, we incubated 8.8 pmol of 40S ribosomal subunits with 90 pmol of IAPV-IRES-STOP RNA in a 15 µl reaction mixture containing Buffer A with 1.7 mM GTP for 5 min at 37°C. Then, the reaction mixture was supplemented with 8.7 pmol of 60S ribosomal subunits and incubated for 5 min at 37°C. Next, we added 27 pmol of eEF2 and 90 pmol eRF* and incubated the reaction for an additional 30 min at 37°C.

### Cryo-EM sample preparation and data acquisition

For ribosomal complexes in a pre-translocated state: 3 µl aliquots of assembled ribosomal complexes at 240–390 nM concentration were incubated for 15 s either on plasma-treated holey carbon grids (QUANTIFOIL R2/2 with homemade continuous carbon film estimated to be 50 Å thick) or on plasma-treated holey gold grids [UltrAuFoil R1.2/1.3 (Russo & Passmore, 2016)]. Grids were blotted for 2.5–3.0 s and flash-cooled in liquid ethane using an FEI Vitrobot. Grids were then transferred to a Polara-G2 microscope operated at 300 kV and equipped with a Gatan K2 Summit direct detector. 11,234 movies of 40 frames were collected in counting mode at 8e⁻/pix/s at a magnification of 31,000 corresponding to a calibrated pixel size of 1.25 Å. Defocus values specified in Leginon (Carragher *et al*, 2000) ranged from 0.8 to 2.5 µm.

For ribosomal complexes in a post-translocated state: 3 µl aliquots of assembled ribosomal complexes around 600 nM were incubated for 15 s on plasma-treated holey gold grids [UltrAuFoil R1.2/1.3 (Russo & Passmore, 2016)]. Grids were blotted for 2.5 s and flash-cooled in liquid ethane using an FEI Vitrobot. Grids were then transferred to a Titan Krios microscope operated at 300 kV and equipped with an energy filter (slits aperture 20 eV) and a Gatan K2 Summit detector. 6,904 movies of 40 frames were collected in counting mode at 8e⁻/pix/s at a magnification of 130,000 corresponding to a calibrated pixel size of 1.0605 Å. Defocus values specified in Leginon (Carragher *et al*, 2000) ranged from 0.5 to 3.0 µm. About half the movies (3,350) were collected at a 35-degree tilt to compensate for preferred orientation that was first identified during screening sessions.

On both microscopes, movies were recorded in automatic mode using the Leginon (Carragher *et al*, 2000) software and frames were aligned using Motioncor2 (Zheng *et al*, 2017). Data collection was monitored and checked on the fly using APPION (Lander *et al*, 2009).

### Image processing and structure determination

For ribosomal complexes in a pre-translocated state, contrast transfer function parameters were estimated using GCTF (Zhang, 2016). For particle picking, a set of templates were generated using density data obtained from a screening session on a F20 microscope that had employed Gaussian picking. Using these templates, particle picking was performed using GAUTOMACH and a particle diameter value of 320 Å. The picked particles were manually screened on the micrographs to remove problematic regions. All 2D and 3D classifications and refinements were performed using RELION (Scheres, 2012). The picked particles were binned four times and subjected to a 2D classification to separate the 40S and 80S particles. We then employed 3D Refine to generate initial

consensus models from both the 40S and 80S particle sets (Appendix Fig S1). We then used our previous CrPV-IRES model (PDB ID 5IT9) to create appropriate masks for these initial models. The mask for the 40S initial model enclosed the putative IRES binding site. The mask for the 80S initial model enclosed the inter-subunit space, the A site finger, the L1 stalk, and ideal helices for SL-III and SL-VI regions of the IAPV-IRES. Using these masks, we performed a round of 3D classification with signal subtraction to remove 3D classes without the IRES. On the classes with the IRES, we performed focused classification without alignment using a new set of masks (Appendix Fig S1). This focused classification step resulted in the final set of classes that were eventually used for modeling.

For ribosomal complexes in a post-translocated state, we employed a similar set of protocols but with a different set of masks that also included regions for eEF2, eRF1*, and the L1 stalk in an extended conformation (Appendix Fig S5).

Final refinements with unbinned data for the selected classed yielded high-resolution maps with density features in agreement with the reported resolution. Local resolution was computed with RESMAP (Kucukelbir *et al*, 2014). New features implemented in Relion 3.0 (Zivanov *et al*, 2018) such as contrast transfer value refinement allowed extending the resolution to close to 3 Å.

### Model building and refinement

Models for the mammalian ribosome and eRF1* were docked into the maps using CHIMERA (Pettersen *et al*, 2004), and COOT (Emsley & Cowtan, 2004) was used to manually adjust the L1 stalk and build the IAPV-IRES using our CrPV-IRES model as initial step. An initial round of refinement was performed in Phenix using real-space refinement with secondary structure restraints (Adams *et al*, 2011). A final step of reciprocal-space refinement using REFMAC was performed (Murshudov *et al*, 1997) for all complexes. The fit of the model to the map density was quantified using FSCaverage and Cref.

## Data availability

Cryo-EM maps for each reconstruction have been deposited in the Electron Microscopy Databank (EMDB) with accession codes EMD-20248; http://www.ebi.ac.uk/pdbe/entry/EMD-20248 (40S-IAPV class 1), EMD-20249; http://www.ebi.ac.uk/pdbe/entry/EMD-20249 (40S-IAPV class 2), EMD-20255; http://www.ebi.ac.uk/pdbe/entry/EMD-20255 (80S-IAPV class 1), EMD-20256; http://www.ebi.ac.uk/pdbe/entry/EMD-20256 (80S-IAPV class 2), EMD-20257; http://www.ebi.ac.uk/pdbe/entry/EMD-20257 (80S-IAPV class 3), and EMD-20258; http://www.ebi.ac.uk/pdbe/entry/EMD-20258 (80S-IAPV-eRF1, post-translocated). Atomic coordinates associated with these maps have been deposited in the Protein Data Bank (PDB) with accession codes 6P4G; http://www.rcsb.org/pdb/explore/explore.do?structureId=6P4G (40S-IAPV class 1), 6P4H; http://www.rcsb.org/pdb/explore/explore.do?structureId=6P4H (40S-IAPV class 2), 6P5I; http://www.rcsb.org/pdb/explore/explore.do?structureId=6P5I (80S-IAPV class 1), 6P5J; http://www.rcsb.org/pdb/explore/explore.do?structureId=6P5J (80S-IAPV class 2), 6P5K; http://www.rcsb.org/pdb/explore/explore.do?structureId=6P5K

(80S-IAPV class 3), and 6P5N; http://www.rcsb.org/pdb/explore/explore.do?structureId=6P5N (80S-IAPV-eRF1, post-translocated).

**Expanded View** for this article is available online.

## Acknowledgements

We are thankful to Vera Pisareva and Andrey Pisarev for a generous donation of ribosomal subunits and eRF1*. We acknowledge Bob Grassucci for technical assistance in data acquisition. Part of this work was performed at the Simons Electron Microscopy Center and National Resource for Automated Molecular Microscopy located at the New York Structural Biology Center, supported by grants from the Simons Foundation (SF349247), NYSTAR, and the NIH National Institute of General Medical Sciences (GM103310). We are especially grateful to Ed Eng, Bill Rice, and Laura Kim for support in data acquisition. Density maps have been deposited at the EMDB with accession codes 20248, 20249, 20255, 20256, 20257, and 20258. Atomic coordinates have been deposited in the PDB with accession codes 6P4G, 6P4H, 6P5I, 6P5J, 6P5K, and 6P5N. J.F. is funded by a National Institutes of Health grant (GM029169).

## Author contributions

ISF conceived the project, designed the experiments, and collected the data with assistance of FA-R and RN. RN produced all RNA constructs with assistance of FA-R. Data processing was carried out by FA-R with RN assistance and ISF supervision. Modeling and model refinement were done by ISF. Data analysis and manuscript drafting were done by ISF with JF assistance. All authors contributed to article writing.

## Conflict of interest

The authors declare that they have no conflict of interest.

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
