## [Review Process File · The EMBO Journal]

The Israeli Acute Paralysis Virus IRES captures host ribosomes by mimicking a ribosomal state with hybrid tRNAs.

Review timeline:

Submission date: 10th April 2019
Editorial Decision: 17th May 2019
Revision received: 11th June 2019
Editorial Decision: 17th July 2019
Revision received: 20th August 2019
Editorial Decision: 30th August 2019
Revision received: 2nd September 2019
Accepted: TBC

Transaction Report:

1st Editorial Decision

17th May 2019

Thank you for submitting your manuscript on the interaction of the Israeli Acute Paralysis Virus IRES with eukaryotic ribosomes for consideration by The EMBO Journal. We have now received three referee reports on your study, which are included below for your information.

As you will see, the reviewers are overall positive and express interest in the study. However, they also raise several concerns that would need to be addressed in a revised version of the manuscript. In addition to more specific technical issues, in particular regarding the resolution of the reported structures, the referees also find that certain conclusions are to some extent speculative and not fully supported by the provided data, for example the aspect of structure-based drug design. Here additional experimental data would need to be provided, or discussed as a potential future application.

Should you be able to adequately address all concerns raised by the referees then we would be happy to consider this study further for publication. I would therefore like to invite you to prepare and submit a revised manuscript.

REFeree REPORTS

Referee #1:

This work presents six high-resolution cryo-EM structures that show how the Israeli Acute Paralysis Virus internal ribosomal entry site (IAPV-IRES) hijacks the eukaryotic ribosome to redirect it towards the production of viral proteins. IAPV has been linked to colony collapse disorder, a syndrome that is responsible for the current worldwide decline in bee populations. Unlike cricavirus IRES, the IAPV-IRES captures host ribosomes by mimicking hybrid tRNA states seen in a pre-translocation translation intermediate. The novel mechanism of IRES action revealed by this study could facilitate the design of antisense RNAs to fight the spread of colony collapse disorder and is thus of general significance and high priority to the wider readership of the EMBO Journal.

Overall, the experiments and analyses are performed to high standards. Results are clearly presented and fully support the conclusions. I therefore only have a few minor comments to make:

- The panels on the right side of Fig. 2E showing A6609 and A1085 are redundant. Keeping a single

panel would be enough.

- Line. 179 - Reference should be to Fig. 3E only.
- In Fig. 3, a label indicating the various domains of the IAPV-IRES would be helpful. Alternatively, a single color key in the figure legend could be used to remind the reader of the color scheme for each domain. The same applies to later figures.
- Lines 210-211: while it is clear that the A, P and E-sites are blocked by the IAPV-IRES, it would be helpful to remind the non-specialist reader which are the key canonical initiation factors that are prevented from binding.
- In Fig. 4B, it would be helpful to label the L1-stalk.
- Lines 244-247. Could the authors please clarify whether the lack of a requirement for eIF5B is physiologically relevant? In other words, is subunit assembly in the absence of eIF5B driven by the concentrations of ribosome and IAPV-IRES used in vitro, which does not necessarily reflect what happens in vivo?
- Figure S1: some indication of the number of particles for the different classes would be useful.
- Defocus ranges shown in Table 1 do not match those in the methods section.

Referee #2:

The manuscript by Acosta-Reyes et al. addresses the structural analysis of a eukaryotic ribosome complex with an IRES RNA from a virus infecting bees, the Israeli Acute Paralysis Virus. The analysis is based on two reconstituted complexes, of which one was designed to comprise a stop codon so that the complex can be stalled in a post-translocation complex using release factor eRF1. This study provides interesting insights into the binding mechanism of the IRES to the ribosome, into the localization of the RNA domains, and into the hijacking and translocation mechanism. However, there are a number of issues and concerns to be addressed before this manuscript can be considered for publication, the most important being overstatements of resolution or oversimplifications that give misleading impressions, either of the number of complexes studied (abstract) or the real resolution level and far how molecular details can really be addressed. Also, the text should be proofread with regards to poor formulations and numerous typos which distract the reader from the main message even at the refereeing stage of this manuscript. Assuming that this is the first structural analysis of an IAPV IRES ribosome complex (not clearly stated), there would be a significant level of novelty in this study that would support publication after an appropriate revision.

Detailed points:

- cryo-EM maps and atomic coordinates should be provided via PDB and EMDB (and EMPIAR) databases.
- abstract: the statement that 6 structures were solved is misleading, in fact 2 complexes were reconstituted (consistently only 2 are shown in the movie) and structural sorting was performed on these; needs rephrasing.
- abstract: resolution statement is clearly overstated and gives a wrong impression at this stage; the maps shown in the figures later clearly show that these are not 3 Å maps. Such resolution values would imply that molecular details of interactions of side-chains can be described, which is clearly not the case. Avoiding overstatements would not diminish the impact of the obtained results, rather it would make it more trustworthy.
- abstract: basing RNAi design on this structure seems unlikely to be realistic, this would be done by high-throughput screening anyway; needs to be toned down.
- introduction, line 37: other countries considered as well?
- various typos to be corrected: an RNA, an IRES, an mRNA etc.
- line 79/80 & 108: concept of mimicking a pre-translocation state is known; should instead be discussed in the final discussion section
- section around lines 90: is somehow related to the topic, but is not conducting to it properly, needs rephrasing
- line 99 & 110: if RNAi has already proved effective against, what is the point of developing RNAi? Contradicts with the statement in the abstract. If this point was really strong, one would expect proposition of complementary RNA segments based on the structure that could interfere with ribosome binding (e.g. in the discussion); otherwise this is giving a wrong impression. In other words, the primary reason to analyse this particular IRES is not RNAi design but instead getting

mechanistic insights which would be an interesting purpose already.

- lines 116, 121, 132, 182 etc. many typos, needs full proof reading before submission.

- line 142: RNA species: how were these identified as being IAPV IRES RNA?

- line 147: "no further purification": in fact there was a sucrose gradient of the complex

- line 150: the statement of "high resolution" is misleading, the map in Fig. 2 barely resolves the helical character of RNA helices, thus this is not high resolution

- Fig. 1A: secondary structure of the IRES: predicted how? any biochemical evidence for this?

Variable loop: how variable, sequence or length? Clarify.

panel B: the gel seems cut at the bottom, the full gel should be shown; legend D: 40S "group": reformulate

- line 156: "mobile": should say "more mobile"

- Fig. 2: why is the 40S subunit rotated in panel B compared to panel A? Better to keep the same orientation

panel C: the IRES helical elements appears to be positioned across all 3 tRNA sites, but this is not a mimicry; needs to be clarified, mainly in the text; important issue because the whole discussion is based on the concept of tRNA mimicry, but in fact the acceptor stem is not mimicked, thus the comparison and statement is an unnecessary overstatement.

Legend, specify: mammalian (rabbit)

Display of unsharpened maps: why? Would sharpening affect the connectivity of densities in the map? If so, it indeed points to the problem discussed above regarding resolution, i.e. it indicates that the map resolution is strongly overestimated. FSC calculations are not a proof of particular resolution values, they need to correspond to the observed features in the map; this could indicate some problems with FSC calculations, masks etc.

- line 190: "follow" implies that at least 2 structural states are found/described to address a movement; possibly rephrase

- line 199: excluding the acceptor stem means that indeed the IRES RNA is not a real tRNA mimic. In other words, there is only one RNA helix left to compare with a tRNA. Furthermore, it needs to be clarified whether that particular helical segment really has the same orientation as a tRNA (in classical or intermediate state), because in the figure the helix appears to stretch over the 3 tRNA sites, which would be perpendicular to the decoding stem of a classical tRNA. Maybe this is only a matter of the figure, but it needs to be presented more clearly.

- Fig. 3: one more illustration that the resolution assessment is overstated, e.g. panels D & E do not resolve individual nucleotide bases as would be expected for a 3.1 Å structure. What is seen here in the map corresponds more to a resolution range of 3.5-4 Å. Again, this does not preclude from addressing the overall localisation of the IRES RNA and its mechanism, but it does mislead the readership to which extent the molecular details can really be addressed.

Legend: alignments excluding IRES and 40S head: were these focused refinement? Same in the methods section, should be clarified;

- Fig. 4: again, resolution issue: no details visible due to limited resolution, in contrast also to statements in the MS and abstract. Why are unsharpened maps shown? Issues with map quality after post-processing point to problems with image processing or incomplete refinement; or/and issues with mask and FSC calculation, see above.

- line 242 etc.: is the IRES helix really in that orientation? In any case, there seems to be only a partial overlap with a classical tRNA, see comments above. The statement of full tRNA mimicry could be misleading, especially when compared to other IRES complexes or tmRNA for example. Mimic of T and D arms of the tRNA: from Fig. 4 it is clear that this is incorrect: a "merge" of 2 tRNAs would be needed for this hypothesis, which does not seem to make sense; clarify and show super-position of T/D arms under an appropriate angle / view to be able to address this.

- Fig. 5, panel C: why do suddenly side-chain densities become visible if this is a zoom of panel B in which only the helicity of the RNA helix can be addressed? Are these filtered versus non-filtered maps? Not stating this clearly could be understood as map manipulation or selective display; to avoid such an impression it would be helpful to in addition show several regions of the map, for example as Suppl. Figures; however, the same map needs to be shown throughout, and at the the same contour level. Densities for IRES and ribosome parts must be displayed at the same contour level.

- line 270: the conclusion is likely to be correct, but it would help to show a comparison with such a conformation (e.g. other known ribosome structures in the presence of initiation factors), to be made and described.

- line 274: "multiple conformational states" is a misleading statement because once bound to the ribosome they in fact adopt discrete, well-defined conformations

- Fig. 6: is eEF2 bound somewhere? If not, the scheme in panel A is misleading with that regards.
- line 306: E-site tRNA is not present in this complex, so why is it discussed?
- repetitive typos on "peptidyl" to be corrected throughout
- section around line 320: lengthy description, known already
- line 348: CCD already introduced; rather unrelated with IRES mechanism discussed in this paper
- line 355: if "putative", what is the basis of the hypothesis of drug design, is it the correct target finally?
- line 374 etc., aparavirus IRES: is this now a general conclusion based on this study or was it already known from aparavirus IRESs?
- Fig. 7: if to compare these IRESs: (secondary) structure of the IRES should be compared/discussed too
- line 398: in contrast to what is stated this is not a high-resolution cryo-EM analysis
- references: numerous typos (80S, not 80s; IRES, not ires etc.) to be corrected (or proper reference software to be used)
- methods: overall appear to be technically fund; however, why was a Polara microscope used for the pre-translocation complex which is the main focus of this study?
- image processing and focused classification and refinement: there are several reviews from different groups in the literature on this method; needs to include a more detailed description of how this was done (masks, regions, partial signal subtraction etc.)

Referee #3:

In this manuscript, the authors have made use of Cryo-EM and single particle analysis to study how the intergenic IRES of the Israeli Acute Paralysis Virus (IAPV) can interact with eukaryotic ribosomes and hijack their host translational machinery. Using two types of in vitro reconstituted complexes, the authors have obtained six near-atomic resolution 3D structures the IAPV-IRES, either associated to the small 40S ribosomal subunit alone, thus corresponding to an early recruitment step, or to « full » 80S ribosomes, in pre- and post-translocated states. The originality of this study is that it describes the first 3D structure of the IAPV intergenic IRES and its interactions with eukaryotic ribosomes, which reveals significant differences with other viral IRES belonging to the type IV family. One of the most striking features of these structures is that the IAPV IRES seem to strongly rigidify the small ribosomal subunit and impede its well characterized rotation/swiveling movements. This work appears well designed and the findings herein proposed are overall in good agreement with the questions asked ; the authors compare their own findings to those of other groups and to other IRES, which makes this study of general interest to the broad readership of the EMBO journal. However, the following comments should be addressed by the authors in order to help the (especially non-specialist) readers to fully grasp the solidity and impact of their study:

- One single 3D structure of the IAPV/ribosome complexes in post-translocated state :
The authors describe both in the text (results and material and methods sections) and in a supplementary figure the very elegant processing scheme that allowed obtaining 5 classes from a huge dataset of reconstituted pre-translocated IRES/ribosomal complexes, which makes their results fairly convincing. In stark contrast, there is little detail, and no figure to help the reader understanding how the authors processed their cryo-EM images of the post-translocated complexes. Why did they choose to describe it so quickly? Was it because only one class featured the IAPV IRES, or only one yielded near-atomic resolution? What about the other particles which were not included in this class? This post-translocated 3D structure would be more convincing with a figure describing the processing scheme that was applied to obtain it.

- Conclusions : Facilitated recruitment of elongation factors, structure-based RNAi design
One of the highlighted findings of this study is that the rigidity of the IAPV-IRES and its tight binding to 80S ribosomes in a pre-translocated complex mimics a ribosomal state with hybrid tRNAs. The authors conclude that this conformation helps to directly recruit translation elongation factors. Are there any published data to sustain this conclusion, or is it a purely (but nonetheless interesting) speculative interpretation of the structural data presented in this manuscript ?
Another example is the iterated assertion that the presented structures will help designing interfering RNA to reduce IAPV disease and bees Colony Collapse Disorder (CCD). Could the authors

illustrate their very strong and exciting conclusions with concrete examples of structure-based design of RNAi, or other drugs? If not, maybe this conclusion might be a bit moderated, and structure-based therapy design presented as longer term solution to fight against bee decline.

- Figures captions :

Again, non-specialist readers might be more convinced by the conclusions reached in this study by a bit more didactics, especially on the figures. For instance, Figures 2, 3, 4, and S2, S3, S4 do not have any captions describing the different domains of the IAPV IRES, or ribosomal domains for figure S2C and S4C.

- Figures numbering is sometimes inverted in the text (for instance, Fig 3D and E panels are cited in the results section before Fig 3C), and some panels are never explicitly cited in the manuscript, which questions their interest.

1st Revision - authors' response

11th June 2019

We are grateful to the reviewers for the in-depth evaluation of the manuscript. Below, we detail the changes introduced in the manuscript and figures and some explanations we feel are necessary in dealing with some comments from reviewer #2. We hope with these changes the manuscript will be considered ready for publication.

Referee #1:

This work presents six high-resolution cryo-EM structures that show how the Israeli Acute Paralysis Virus internal ribosomal entry site (IAPV-IRES) hijacks the eukaryotic ribosome to redirect it towards the production of viral proteins. IAPV has been linked to colony collapse disorder, a syndrome that is responsible for the current worldwide decline in bee populations. Unlike cricavirus IRES, the IAPV-IRES captures host ribosomes by mimicking hybrid tRNA states seen in a pre-translocation translation intermediate. The novel mechanism of IRES action revealed by this study could facilitate the design of antisense RNAs to fight the spread of colony collapse disorder and is thus of general significance and high priority to the wider readership of the EMBO Journal.

Overall, the experiments and analyses are performed to high standards. Results are clearly presented and fully support the conclusions. I therefore only have a few minor comments to make:

- The panels on the right side of Fig. 2E showing A6609 and A1085 are redundant. Keeping a single panel would be enough.

-We have modified Fig. 2E as suggested.

- Line. 179 - Reference should be to Fig. 3E only.

-We have corrected this.

- In Fig. 3, a label indicating the various domains of the IAPV-IRES would be helpful.

Alternatively, a single color key in the figure legend could be used to remind the reader of the color scheme for each domain. The same applies to later figures.

-All figures now show labels for the IAPV-IRES domains in a common color key specified in the figure legend.

- Lines 210-211: while it is clear that the A, P and E-sites are blocked by the IAPV-IRES, it would be helpful to remind the non-specialist reader which are the key canonical initiation factors that are prevented from binding.

- We have modified the sentence citing initiation factors eIF1, eIF1A and eIF5B which are the ones whose binding sites overlap with the IAPV-IRES binding site.

- In Fig. 4B, it would be helpful to label the L1-stalk.

-A label for the L1-stalk in panel A and B has been included.

- Lines 244-247. Could the authors please clarify whether the lack of a requirement for eIF5B is physiologically relevant? In other words, is subunit assembly in the absence of eIF5B driven by the concentrations of ribosome and IAPV-IRES used in vitro, which does not necessarily reflect what happens in vivo?

- As stated above, the binding of IAPV-IRES to the 80S ribosome is incompatible with the simultaneous binding of eIF5B which unambiguously indicates that, in physiological conditions, IRES-driven subunit joining must proceed without the involvement of eIF5B.
- Figure S1: some indication of the number of particles for the different classes would be useful.
- Percentages indicating the relative distribution of particles after classification are now shown. The specific number of particles for each class is reported in Table S1.
- Defocus ranges shown in Table 1 do not match those in the methods section.
- We are grateful to the reviewer for noticing this. In the methods section the defocus range reported correspond to the range specified for data collection. In table S1 we report the actual values calculated after ctf estimations. Clarification of this point is now provided at the figure legend of Table S1 as well as in the methods section.

Referee #2:

The manuscript by Acosta-Reyes et al. addresses the structural analysis of a eukaryotic ribosome complex with an IRES RNA from a virus infecting bees, the Israeli Acute Paralysis Virus. The analysis is based on two reconstituted complexes, of which one was designed to comprise a stop codon so that the complex can be stalled in a post-translocation complex using release factor eRF1. This study provides interesting insights into the binding mechanism of the IRES to the ribosome, into the localization of the RNA domains, and into the hijacking and translocation mechanism. However, there are a number of issues and concerns to be addressed before this manuscript can be considered for publication, the most important being overstatements of resolution or oversimplifications that give misleading impressions, either of the number of complexes studied (abstract) or the real resolution level and far how molecular details can really be addressed. Also, the text should be proofread with regards to poor formulations and numerous typos which distract the reader from the main message even at the refereeing stage of this manuscript. Assuming that this is the first structural analysis of an IAPV IRES ribosome complex (not clearly stated), there would be a significant level of novelty in this study that would support publication after an appropriate revision.

Detailed points:

- cryo-EM maps and atomic coordinates should be provided via PDB and EMDB (and EMPIAR) databases.
- PDB ID and EMDB ID are now reported as the data has been deposited.
- abstract: the statement that 6 structures were solved is misleading, in fact 2 complexes were reconstituted (consistently only 2 are shown in the movie) and structural sorting was performed on these; needs rephrasing.
- We respectfully disagree with the reviewer on this point. There is a difference between what the reviewer considered a "reconstituted" complex and what we actually designed as an "*in vitro* reconstituted reaction". Our *in vitro* reactions were designed to contain several complexes able to be distinguished by *in silico* classification methods. Thus, the statement "6 structures were solved" is indeed correct. Nevertheless, we rephrased the abstract to make clear that two reconstitution reactions were imaged, which in total yielded 6 structures.
- abstract: resolution statement is clearly overstated and gives a wrong impression at this stage; the maps shown in the figures later clearly show that these are not 3 Å maps. Such resolution values would imply that molecular details of interactions of side-chains can be described, which is clearly not the case. Avoiding overstatements would not diminish the impact of the obtained results, rather it would make it more trustworthy.
- In this comment as well as several below, the reviewer questions the resolutions reported for the structures presented. We have rigorously followed the current standards in the field, which involves the processing of cryoEM data using independent half-maps in order to avoid overfitting of the data. When following this approach, resolution estimation following the 0.143 cutoff of FSC between half maps is a well-established criterion in resolution estimation, and indeed our maps present features expected from that estimated resolution. To illustrate this, we have now included an additional supplementary figure (Figure S4, A and B) with a slice through one of our maps where it can be appreciated that density features expected from a 3 Å map appear: clear individual side chains in proteins and base separation in nucleic acids. The resolution of our maps is thus not over-estimated.

However, the resolution in our maps is not homogeneous, and specific regions deviate from the nominal resolution value reported. This is not uncommon in maps of large biological entities and it is actually very normal in ribosome maps with large ligands as protein factors and/or IRESs. In such situations, the accepted practice in the field is reporting local resolution maps, normally presented as colored unsharpened maps indicating the resolution across the entire map. We have done this for every reconstruction presented, so the reader can judge which regions of the IRES exhibit a lower resolution and which ones a higher one (Figures S2, S3 and S6). Segments of the IAPV-IRES stabilized by ribosomal elements exhibit a higher resolution while those not stabilized or in contact with intrinsic mobile ribosomal elements deviate from the nominal resolution. We feel the best way to display the continuity of the density of areas of the maps with variable resolutions is through the use of unsharpened maps, especially if a large region of the map is depicted. In those areas where the maps are improved, we show the maps post-processed with B factors indicated in Table S1. In order to accommodate the referee demands, we have modified the abstract, removing the reference to the nominal (overall) resolution of the maps and we have also added a paragraph at the beginning of the results section where it is clearly stated that the resolution of the IRES is variable and the reader is referred to the relevant supplementary figures where maps colored according to the local resolution can be found. We have also modified the figure legends to clearly specify where unsharpened maps are shown and where post-processed maps.

- abstract: basing RNAi design on this structure seems unlikely to be realistic, this would be done by high-throughput screening anyway; needs to be toned down.
- We believe our suggestion is realistic as current RNAi experiments are actually effective. The structures presented will allow a more precise design of the anti-sense RNA molecules. In any case, we have modified the abstract toning down this aspect.
- introduction, line 37: other countries considered as well?
- No data are available apart from the US
- various typos to be corrected: an RNA, an IRES, an mRNA etc.
- Corrected.
- line 79/80 & 108: concept of mimicking a pre-translocation state is known; should instead be discussed in the final discussion section
- We respectfully disagree with this comment as the concept of pre-translocation mimicry has to be introduced before discussing the diversity of the type IV family of IRESs.
- section around lines 90: is somehow related to the topic, but is not conducting to it properly, needs rephrasing
- Line 90 describes the previously reported cryoEM structure of the TVS-IRES in complex with the yeast ribosome which is directly related to the main topic of the present manuscript. We comment on this work as is essential for the reader to know which is the current status of structural biology regarding IRES/ribosome complexes.
- line 99 & 110: if RNAi has already proved effective against, what is the point of developing RNAi? Contradicts with the statement in the abstract. If this point was really strong, one would expect proposition of complementary RNA segments based on the structure that could interfere with ribosome binding (e.g. in the discussion); otherwise this is giving a wrong impression. In other words, the primary reason to analyse this particular IRES is not RNAi design but instead getting mechanistic insights which would be an interesting purpose already.
- We believe there is much room for improvement in RNAi technology against CCD. Proof of principle pioneering studies reported the feasibility of this strategy, however, the lack of understanding in structural terms on how the IAPV-IRES works, precluded further fine-tuning of the designed RNAi. We believe the structural framework here presented may significantly impact the applicability of this technology in the fight against CCD. We see as complementary aspects of this study its general interest from a purely biological perspective and the possibility of improving RNAi design through the structures presented.
- lines 116, 121, 132, 182 etc. many typos, needs full proof reading before submission.
- Corrected.
- line 142: RNA species: how were these identified as being IAPV IRES RNA?
- The plasmid DNA encoding the RNA fragment transcribed was sequenced and confirmed to be the IAPV-IRES. Purified RNA was mixed with purified ribosomal subunit and the reaction mix analyzed by sucrose gradient. After RNA extraction a band of similar size of IAPV-IRES could be seen in the 40S/80S peak which is absent if the purified RNA is not included in the mix. Ultimately,

cryoEM maps at low and high resolution confirm the presence of the IRES in the complexes.

- line 147: "no further purification": in fact there was a sucrose gradient of the complex
 - The sucrose gradient was used ONLY for analytical purposes. As stated in the text, the final complexes were imaged without sucrose gradient step *id est* "no further purification" after the *in vitro* reaction assembly.

- line 150: the statement of "high resolution" is misleading, the map in Fig. 2 barely resolves the helical character of RNA helices, thus this is not high resolution

-See note above.

- Fig. 1A: secondary structure of the IRES: predicted how? any biochemical evidence for this? Variable loop: how variable, sequence or length? Clarify.

-There is abundance literature about IGR IRESs where secondary structure was predicted using sequence conservation as well as directed mutagenesis. Relevant references are properly cited.

panel B: the gel seems cut at the bottom, the full gel should be shown; legend D: 40S "group": reformulate

- Full gel show now in Figure S4C.

- line 156: "mobile": should say "more mobile"

-Corrected.

- Fig. 2: why is the 40S subunit rotated in panel B compared to panel A? Better to keep the same orientation

- Given that small subunit rotation movements are orthogonal to head swiveling movements, there is only a few orientations where those two movements can be shown in two dimensions. We decided to change the orientation so an easy 90-degree rotation can be performed yielding a front view of the head from a side view of the whole small subunit. We feel this orientation is best to show those movements, especially for the non-specialist reader.

panel C: the IRES helical elements appears to be positioned across all 3 tRNA sites, but this is not a mimicry; needs to be clarified, mainly in the text; important issue because the whole discussion is based on the concept of tRNA mimicry, but in fact the acceptor stem is not mimicked, thus the comparison and statement is an unnecessary overstatement.

Legend, specify: mammalian (rabbit)

- We have added supplementary figure S7 where the mimicry to the tRNA in the A site can be appreciated in several orientations. Also, main features of tRNA structure are indicated as reference. The PKI of the IAPV-IRES overlaps with the position occupied by the anticodon arm, D-arm and T ψ C arms of the tRNA in a A/P hybrid configuration.

Display of unsharpened maps: why? Would sharpening affect the connectivity of densities in the map? If so, it indeed points to the problem discussed above regarding resolution, i.e. it indicates that the map resolution is strongly overestimated. FSC calculations are not a proof of particular resolution values, they need to correspond to the observed features in the map; this could indicate some problems with FSC calculations, masks etc.

-See note above.

- line 190: "follow" implies that at least 2 structural states are found/described to address a movement; possibly rephrase

- We do not understand this comment as indeed 2 structures of the 40S/IAPV-IRES are presented.

- line 199: excluding the acceptor stem means that indeed the IRES RNA is not a real tRNA mimic. In other words, there is only one RNA helix left to compare with a tRNA. Furthermore, it needs to be clarified whether that particular helical segment really has the same orientation as a tRNA (in classical or intermediate state), because in the figure the helix appears to stretch over the 3 tRNA sites, which would be perpendicular to the decoding stem of a classical tRNA. Maybe this is only a matter of the figure, but it needs to be presented more clearly.

-See note above.

- Fig. 3: one more illustration that the resolution assessment is overstated, e.g. panels D & E do not resolve individual nucleotide bases as would be expected for a 3.1 Å structure. What is seen here in the map corresponds more to a resolution range of 3.5-4 Å. Again, this does not preclude from addressing the overall localisation of the IRES RNA and its mechanism, but it does mislead the readership to which extent the molecular details can really be addressed.

-See note above.

Legend: alignments excluding IRES and 40S head: were these focused refinement? Same in the methods section, should be clarified;

-The reviewer is confusing cryoEM particle alignments with alignments between PDB models. We have rephrased the sentence to make this evident.

- Fig. 4: again, resolution issue: no details visible due to limited resolution, in contrast also to

statements in the MS and abstract. Why are unsharpened maps shown? Issues with map quality after post-processing point to problems with image processing or incomplete refinement; or/and issues with mask and FSC calculation, see above.

-See note above.

- line 242 etc.: is the IRES helix really in that orientation? In any case, there seems to be only a partial overlap with a classical tRNA, see comments above. The statement of full tRNA mimicry could be misleading, especially when compared to other IRES complexes or tmRNA for example. Mimic of T and D arms of the tRNA: from Fig. 4 it is clear that this is incorrect: a "merge" of 2 tRNAs would be needed for this hypothesis, which does not seem to make sense; clarify and show super-position of T/D arms under an appropriate angle / view to be able to address this.

-We specify in the text that the mimicry is restricted to specific segments of canonical tRNAs. To make this more evident we have added a supplementary figure (Fig. S7) where the PKI is displayed in different orientations superposed to a canonical tRNA and hybrid A/P-tRNA. We also added in the figure a tRNA diagram with indications of the different components so the reader can quickly identify which segments overlaps with the PKI of the IRES.

- Fig. 5, panel C: why do suddenly side-chain densities become visible if this is a zoom of panel B in which only the helicity of the RNA helix can be addressed? Are these filtered versus non-filtered maps? Not stating this clearly could be understood as map manipulation or selective display; to avoid such an impression it would be helpful to in addition show several regions of the map, for example as Suppl. Figures; however, the same map needs to be shown throughout, and at the the same contour level. Densities for IRES and ribosome parts must be displayed at the same contour level.

-See note above.

- line 270: the conclusion is likely to be correct, but it would help to show a comparison with such a conformation (e.g. other known ribosome structures in the presence of initiation factors), to be made and described.

-The reviewer is confusing an initiation step with a pre-translocation step. Type IV IRES do not mimic initiation stages but specifically a translocation state after initiation. We believe the inclusion of a figure with initiation factors will be confusing for the general readership. Figure 4 already shows superpositions of the IAPV-IRES with pre-translocation structures in canonical and rotated configurations.

- line 274: "multiple conformational states" is a misleading statement because once bound to the ribosome they in fact adopt discrete, well-defined conformations

-This is not correct. It is very well established in the field that ribosomes can populate multiple states with the same ligands bound.

- Fig. 6: is eEF2 bound somewhere? If not, the scheme in panel A is misleading with that regards.

-eEF2 is required for an early translocation step of the IRES before eRF1 binding. Even though is not part of the final complex, early eEF2-catalized IRES translocation is required for an efficient binding of eRF1. Eliminating it from the scheme would be misleading as it could be wrongly assumed that eRF1 can directly bind the ribosome in the presence of just IAPV-IRES.

- line 306: E-site tRNA is not present in this complex, so why is it discussed?

-The whole paragraph introduces tRNA translocation through the ribosome. E-site tRNAs are assisted by the L1-stalk on their way out of the ribosome in a very important phase of translocation. The IAPV-IRES strongly interacts with the L1-stalk and the comparison with a canonical E-site tRNA is mandatory for a full understanding of the IRES mechanism.

- repetitive typos on "peptidyl" to be corrected throughout

-Corrected

- section around line 320: lengthy description, known already

- We have simplified the sentence.

- line 348: CCD already introduced; rather unrelated with IRES mechanism discussed in this paper

-We have deleted this paragraph.

- line 355: if "putative", what is the basis of the hypothesis of drug design, is it the correct target finally?

-CCD is a multifaceted syndrome whose exact etiology is very complex and partially unknown. We feel is best to tone down the importance of IAPV in causing CCD.

- line 374 etc., aparavirus IRES: is this now a general conclusion based on this study or was it already known from aparavirus IRESs?

- It is a general conclusion based on this study.
- Fig. 7: if to compare these IRESs: (secondary) structure of the IRES should be compared/discussed too
- Relevant literature in the field regarding these comparisons is properly cited.
- line 398: in contrast to what is stated this is not a high-resolution cryo-EM analysis
- See note above.
- references: numerous typos (80S, not 80s; IRES, not ires etc.) to be corrected (or proper reference software to be used)
- Corrected.
- methods: overall appear to be technically fund; however, why was a Polara microscope used for the pre-translocation complex which is the main focus of this study?
- The authors do not understand this comment. The Polara microscope is a high-end 300Kev microscope able to deliver very high-resolution reconstructions. We do not share the reviewer's opinion that the use of the Polara microscope could be considered "not technically fund".
- image processing and focused classification and refinement: there are several reviews from different groups in the literature on this method; needs to include a more detailed description of how this was done (masks, regions, partial signal subtraction etc.)
- The senior authors of this manuscripts are among the first in the field to use masked classification in image processing of ribosome samples. The original papers where this technique was first used are cited and we feel that is the correct citation source and not posterior reviews. Additionally, technical details are described in the methods section.

Referee #3:

In this manuscript, the authors have made use of Cryo-EM and single particle analysis to study how the intergenic IRES of the Israeli Acute Paralysis Virus (IAPV) can interact with eukaryotic ribosomes and hijack their host translational machinery. Using two types of in vitro reconstituted complexes, the authors have obtained six near-atomic resolution 3D structures the IAPV-IRES, either associated to the small 40S ribosomal subunit alone, thus corresponding to an early recruitment step, or to « full » 80S ribosomes, in pre- and post-translocated states. The originality of this study is that it describes the first 3D structure of the IAPV intergenic IRES and its interactions with eukaryotic ribosomes, which reveals significant differences with other viral IRES belonging to the type IV family. One of the most striking features of these structures is that the IAPV IRES seem to strongly rigidify the small ribosomal subunit and impede its well characterized rotation/swiveling movements. This work appears well designed and the findings herein proposed are overall in good agreement with the questions asked ; the authors compare their own findings to those of other groups and to other IRES, which makes this study of general interest to the broad readership of the EMBO journal. However, the following comments should be addressed by the authors in order to help the (especially non-specialist) readers to fully grasp the solidity and impact of their study:

- One single 3D structure of the IAPV/ribosome complexes in post-translocated state : The authors describe both in the text (results and material and methods sections) and in a supplementary figure the very elegant processing scheme that allowed obtaining 5 classes from a huge dataset of reconstituted pre-translocated IRES/ribosomal complexes, which makes their results fairly convincing. In stark contrast, there is little detail, and no figure to help the reader understanding how the authors processed their cryo-EM images of the post-translocated complexes. Why did they choose to describe it so quickly? Was it because only one class featured the IAPV IRES, or only one yielded near-atomic resolution? What about the other particles which were not included in this class? This post-translocated 3D structure would be more convincing with a figure describing the processing scheme that was applied to obtain it.

-We are grateful for this suggestion, a new supplementary figure (figure S5) with the classification scheme followed for the post-translocation reaction has been added. Percentage of particles assigned to different classes found are also indicated.

- Conclusions : Facilitated recruitment of elongation factors, structure-based RNAi design
One of the highlighted findings of this study is that the rigidity of the IAPV-IRES and its tight binding to 80S ribosomes in a pre-translocated complex mimics a ribosomal state with hybrid

tRNAs. The authors conclude that this conformation helps to directly recruit translation elongation factors. Are there any published data to sustain this conclusion, or is it a purely (but nonetheless interesting) speculative interpretation of the structural data presented in this manuscript ?

-There is no specific data regarding this issue. As such, we toned down this statement and unambiguously indicate that is an informed guess derived from the structures.

Another example is the iterated assertion that the presented structures will help designing interfering RNA to reduce IAPV disease and bees Colony Collapse Disorder (CCD). Could the authors illustrate their very strong and exciting conclusions with concrete examples of structure-based design of RNAi, or other drugs? If not, maybe this conclusion might be a bit moderated, and structure-based therapy design presented as longer term solution to fight against bee decline.

-There is a study where bee hives affected by CCD syndrome were feed with antisense RNAs directed toward the IAPV-IRES and that strategy proved successful in preventing further deterioration of the colony. With this in mind, we believe our structures will assist on localizing better areas for the design of such anti-sense RNAs in the IAPV-IRES.

- Figures captions :

Again, non-specialist readers might be more convinced by the conclusions reached in this study by a bit more didactics, especially on the figures. For instance, Figures 2, 3, 4, and S2, S3, S4 do not have any captions describing the different domains of the IAPV IRES, or ribosomal domains for figure S2C and S4C.

- Colored labels for the different IRES domains have been added to the figures.

- Figures numbering is sometimes inverted in the text (for instance, Fig 3D and E panels are cited in the results section before Fig 3C), and some panels are never explicitly cited in the manuscript, which questions their interest.

-We have corrected this.

2nd Editorial Decision

17th July 2019

Thank you for submitting your revised manuscript for our consideration. It has now been seen once more by the original referees (see comments below). As you will see the referees find that their comments have largely been addressed. However referee #2 still raises some specific points, which should be addressed, and will require textual changes, as well as including additional map versions, and updating the EMDB/PDB depositions in a revised final version. We shall therefore be happy to accept the study for publication after a final minor revision to address these issues.

 REFEREE REPORTS.

Referee #1:

I thank the authors for providing a revised version of their manuscript. In my opinion, all of the comments raised during the initial review have now been addressed in a satisfactory manner and I have no further concerns or suggestions to report.

Referee #2:

The revised version of the manuscript is now much improved, even though not finalized yet (see comments below). Specifically, it now clarifies the confusing issue of using non-filtered and filtered maps in the same figure (at least for most of them apart from one, see below). While this remains initially misleading when looking at the figures the legend now clarifies this better. The post-processed map of that same region should be shown in the Suppl. Mat.

Fig. 3: maybe a slightly higher contour level should be used.

Fig. 5: the zoom drawing still is confusing, here zooming + postprocessing/filtering is performed (which is counter-intuitive to the reader). In addition, it would be good to show the maps of the same regions of all these figures in the post-processed version in the Suppl. Mat.

Suppl. S4: specify whether filtered or non-filtered maps are shown. Show maps also for the less well defined regions of the IRES RNA, specify that this is without post-processing/filtering.

Regarding image processing, while the authors in the past undoubtedly also worked on issues related with heterogeneity, they were not the only ones. There would be no harm for them to mention publications from 2 or 3 other laboratories, including the original papers or recent reviews on local & focused classifications and refinements.

Furthermore, now that I had a chance to look at the structure based on the coordinate and cryo-EM map (filtered and non-filtered) files provided by the authors, it does confirm that the specific parts of interest, i.e. the IRES, is not well defined in the maps. This should be clearly written from the beginning of the manuscript to avoid confusing the reader. This in fact explains why the unfiltered maps are shown in the figures, because the post-processed/filtered maps are much too noisy for the IRES RNA. In fact, both filtered and unfiltered (or moderately filtered) maps should be deposited in the EMDB. It is important to make sure that the map is shown in the same manner for the ribosome and the IRES regions in the various figures (in the case these are different it would be helpful to indicate the contour level (rmsd) for each figure panel).

One possibility to improve the interpretation of the maps could be to do local filtering taking into account the B-factors of the atomic model. In fact, the atomic model needs further refinement: amino acid side chains and their conformations are not well fitted into the density, and the geometry of the RNA is rather unreliable in some places of the structure (particularly for the IRES RNA). The final refined atomic model will need to be updated for the EMDB / PDB deposition.

Referee #3:

In this revised version of the manuscript, the authors made great efforts to better describe their results, with for instance more detailed and didactic figures, and a more precise depiction of the image analysis techniques employed. Furthermore, they also toned down some of their boldest statements, so the reader can now distinguish between what is a true conclusion and what might be an interpretation derived from the findings therein presented.

In my opinion, the presentation of this work has been significantly improved and appears now really strong and convincing. All my previous comments (which were mostly about the form of the manuscript) were judiciously addressed, I thus think that this article is now worth publishing in the EMBO Journal.

2nd Revision - authors' response

20th August 2019

We have introduced the changes specified below in order to accommodate the majority of reviewer #2 requests. We would like to also remind the editor that all maps, masks and models are deposited in publicly accessible databases, so any reader can have a direct look at the maps and models at any moment, making some of the reviewer #2 requests really redundant. We really hope with these changes the manuscript will be considered ready for publication.

The revised version of the manuscript is now much improved, even though not finalized yet (see comments below). Specifically, it now clarifies the confusing issue of using non-filtered and filtered maps in the same figure (at least for most of them apart from one, see below). While this remains initially misleading when looking at the figures the legend now clarifies this better. The post-processed map of that same region should be shown in the Suppl. Mat.

Fig. 3: maybe a slightly higher contour level should be used.

This has been done.

Fig. 5: the zoom drawing still is confusing, here zooming + postprocessing/filtering is performed (which is counter-intuitive to the reader). In addition, it would be good to show the maps of the same regions of all these figures in the post-processed version in the Suppl. Mat.

The unsharpened density for the large panel has been removed and the molecular model only is shown now. The zoomed view of the model inserted in the final post-processed map is the only density shown in the figure now.

Suppl. S4: specify whether filtered or non-filtered maps are shown. Show maps also for the less well defined regions of the IRES RNA, specify that this is without post-processing/filtering.

It is now explicitly specified in the figure that the map is post-processed.

Regarding image processing, while the authors in the past undoubtedly also worked on issues related with heterogeneity, they were not the only ones. There would be no harm for them to mention publications from 2 or 3 other laboratories, including the original papers or recent reviews on local & focused classifications and refinements.

A reference for a review on local and focused classifications from another group is now included.

Furthermore, now that I had a chance to look at the structure based on the coordinate and cryo-EM map (filtered and non-filtered) files provided by the authors, it does confirm that the specific parts of interest, i.e. the IRES, is not well defined in the maps. This should be clearly written from the beginning of the manuscript to avoid confusing the reader.

This paragraph at the beginning of the results section addresses this concern:

" The nominal resolution of the maps was calculated to be around 3 Å (Appendix Fig. S2, S3 and S6). In the best areas, such as the 60S subunit and the body of the 40S subunit, the maps exhibit characteristics in accordance with this resolution, with very well resolved side chains in proteins and clear base separation in the ribosomal RNA components (Appendix Fig. S4A and B). However, the resolution of the IRES density, due to the intrinsic flexibility of this component, deviates from the nominal resolution. Areas of the IRES stabilized by ribosomal components are well resolved, with local resolution better than 4 Å (Fig. 3D and E), while areas not stabilized by the ribosome or in contact with intrinsically dynamic elements of the ribosome like the L1-stalk, exhibit lower local resolution (Appendix Fig. S2, S3 and S6). In order to properly visualize the continuity of the maps for the full IRES, we show the unsharpened maps in the figures, especially where large areas of the maps are depicted. In those regions of the maps exhibiting resolution better than 4 Å for the IRES, maps sharpened with B factors reported in Table S1 are shown."

This in fact explains why the unfiltered maps are shown in the figures, because the post-processed/filtered maps are much too noisy for the IRES RNA. In fact, both filtered and unfiltered (or moderately filtered) maps should be deposited in the EMDB. It is important to make sure that the map is shown in the same manner for the ribosome and the IRES regions in the various figures (in the case these are different it would be helpful to indicate the contour level (rmsd) for each figure panel.

Once again, all maps, masks and models are deposited in public databases, so any reader can download and generate the map he/she wishes with a specific B-factor.

One possibility to improve the interpretation of the maps could be to do local filtering taking into account the B-factors of the atomic model. In fact, the atomic model needs further refinement: amino acid side chains and their conformations are not well fitted into the density, and the geometry of the RNA is rather unreliable in some places of the structure (particularly for the IRES RNA). The final refined atomic model will need to be updated for the EMDB / PDB deposition.

This has been done.

3rd Editorial Decision

30th August 2019

Thank you for submitting your revised manuscript for our consideration. I am pleased to say we will be happy to formally accept the study for publication after some final minor editorial issues that are listed in detail below are addressed.

Corresponding Author Name: Joachim Frank and Israel S. Fernandez

Manuscript Number: EMBOJ-2019-102226R